# A forward genetic screen reveals novel independent regulators of ULBP1, an activating ligand for natural killer cells

Benjamin G Gowen[1], Bryan Chim[2], Caleb D Marceau[3], Trever T Greene[1], Patrick Burr[2], Jeanmarie R Gonzalez[1], Charles R Hesser[1], Peter A Dietzen[1], Teal Russell[1], Alexandre Iannello[1], Laurent Coscoy[1], Charles L Sentman[4], Jan E Carette[3], Stefan A Muljo[2], David H Raulet[1]*

[1]Department of Molecular and Cell Biology, Cancer Research Laboratory, University of California, Berkeley, Berkeley, United States; [2]Laboratory of Immunology, National Institute of Allergy and Infectious Diseases, Bethesda, United States; [3]Department of Microbiology and Immunology, Stanford University School of Medicine, Stanford, United States; [4]Center for Synthetic Immunity, Department of Microbiology and Immunology, Dartmouth Geisel School of Medicine, Lebanon, United States

**Abstract** Recognition and elimination of tumor cells by the immune system is crucial for limiting tumor growth. Natural killer (NK) cells become activated when the receptor NKG2D is engaged by ligands that are frequently upregulated in primary tumors and on cancer cell lines. However, the molecular mechanisms driving NKG2D ligand expression on tumor cells are not well defined. Using a forward genetic screen in a tumor-derived human cell line, we identified several novel factors supporting expression of the NKG2D ligand ULBP1. Our results show stepwise contributions of independent pathways working at multiple stages of ULBP1 biogenesis. Deeper investigation of selected hits from the screen showed that the transcription factor ATF4 drives *ULBP1* gene expression in cancer cell lines, while the RNA-binding protein RBM4 supports ULBP1 expression by suppressing a novel alternatively spliced isoform of *ULBP1* mRNA. These findings offer insight into the stress pathways that alert the immune system to danger.

*For correspondence:
raulet@berkeley.edu

**Competing interests:** The authors declare that no competing interests exist.

## Introduction

Natural killer (NK) cells are lymphocytes of the innate immune system that play a critical role in limiting tumor growth (*Vivier et al., 2011*; *Marcus et al., 2014*; *Mittal et al., 2014*). NK cell activation is controlled by a balance of signals from activating and inhibitory receptors, which recognize cognate ligands expressed by potential target cells (*Vivier et al., 2011*; *Shifrin et al., 2014*). One of the best-studied NK-activating receptors is NKG2D, which is also expressed on certain subsets of T cells (*Raulet, 2003*). Engagement of NKG2D by its ligands displayed on a target cell membrane leads to NK cell activation, cytokine secretion, and lysis of target cells, such as tumor cells.

NKG2D recognizes a family of ligands that are structurally similar to MHC Class I proteins. Humans express up to eight NKG2D ligands (ULBP1-6, MICA, and MICB), and mice express 5–6 different ligands, depending on the strain (RAE-1α-ε, H60a-c, and MULT1) (*Raulet et al., 2013*). Healthy cells typically do not display NKG2D ligands on their surface and are thus poor targets for NKG2D-mediated lysis by NK cells. However, cellular stresses associated with transformation, viral infection, or other danger to the host cause the upregulation of NKG2D ligand expression (*Raulet et al., 2013*). Primary tumors and cancer cell lines frequently express one or more NKG2D ligand, and NKG2D expression is important for the control of tumors in vivo in models of spontaneous cancer (*Guerra et al., 2008*).

**eLife digest** Cancer is caused by a series of mutations that result in uncontrolled cell growth and division. Yet, the body's immune system can often detect and destroy abnormal cells before they cause tumors and disease. Natural killer cells are part of the immune system and have receptors on their surface that allow them to tell the difference between healthy host cells and host cells that are stressed or abnormal. Some of these receptors activate the natural killer cells when they bind to their target molecules. Other receptors have the opposite effect and inhibit the natural killer cells. Activation occurs when the signaling from the activating receptors is stronger than the signals from the inhibitory receptors.

One of the well-studied activating receptors recognizes a number of proteins and molecules that are produced by abnormal or tumor cells, including a protein called ULBP1. This protein is absent from the surface of healthy cells but is found in abundance on tumor cells. However, it is still not clear what drives tumor cells to produce ULBP1 (or other molecules) that are recognized by natural killer cell receptors.

Now, Gowen et al. report on a genetic screen that has revealed numerous genes that regulate the levels of ULBP1 in human cells. Many of these genes had independent effects that when added together accounted for most of the ULBP1 present on the cell surface.

Gowen et al. then explored some of the 'regulators' encoded by these genes in more detail. One called ATF4, which had previously been linked to stress responses, was shown to increase the expression of the gene for ULBP1 in cancer cells. Another regulator called RBM4 instead acted in a different way and at a later stage in ULBP1 production.

All together, these findings offer insight into the stress pathways that alert the immune system to abnormal cells. The next challenge will be investigating how these pathways might be exploited for cancer immunotherapy.

Tumors arise despite the tumor-suppressive effects of the immune system, and some tumors show evidence of adaptation to escape immune control (*Schreiber et al., 2011*). In the case of NKG2D-mediated tumor recognition, published results suggest that one mechanism of tumor immune evasion is the loss or decreased expression of NKG2D ligands (*Guerra et al., 2008*; *McGilvray et al., 2009*). In other cases, tumors progress despite sustained expression of NKG2D ligands (*Vetter et al., 2002*; *Guerra et al., 2008*; *McGilvray et al., 2009*; *Hilpert et al., 2012*). The evidence as a whole suggests that upregulation of NKG2D ligands on early stage tumor cells is part of a host defense mechanism, but that the immune response subsequently applies selective pressure for tumors that have either extinguished expression of NKG2D ligands or have activated immune suppressive mechanisms (*Raulet and Guerra, 2009*). Therefore, identifying factors and pathways that regulate NKG2D ligands will improve our understanding of the cellular properties used by the immune system to define unwanted cells and will also help reveal how tumors evade the corresponding immune responses.

Prior investigations have identified regulators of NKG2D ligands using a candidate approach based on the roles of these regulators in known stress pathways. Such approaches have implicated the DNA damage response pathway (*Gasser et al., 2005*), heat shock (*Venkataraman et al., 2007*; *Nice et al., 2009*), hyperproliferation (*Jung et al., 2012*), and pattern recognition receptors (*Hamerman et al., 2004*), among others, in the regulation of one or more NKG2D ligands. However, these pathways do not account fully for expression of ligands in tumor cells, since inhibiting them may decrease ligand expression but typically does not abrogate it. For example, the DNA damage response is active in many cancer cells and tumor cell lines, but inhibiting that pathway only partially inhibits ligand expression (*Gasser et al., 2005*; *Gasser and Raulet, 2006*; *Soriani et al., 2014*). Similarly, hyperproliferation can drive NKG2D ligand expression, but blocking proliferation does not completely eliminate NKG2D ligand expression by tumor cell lines (*Jung et al., 2012*). These findings suggest that unidentified molecular cues in tumor cells also initiate the expression of NKG2D ligands, allowing potentially dangerous tumor cells to be distinguished from normal cells. Identifying those cues, especially for human NKG2D ligands, is important for understanding the biological regulation of NKG2D ligands and devising approaches for immunotherapy based on that knowledge.

To identify novel drivers of NKG2D ligand expression, we performed a genome-wide loss-of-expression mutant screen in the tumor-derived human cell line HAP1 (*Carette et al., 2009*; *Carette et al., 2011*) and used CRISPR/Cas9 gene targeting methodology for confirmation of the hits and extension of the results. The results reveal previously unknown regulators for NKG2D ligands, provide evidence for selectivity of the regulators for specific ligands, and support the cooperation of different stress pathways in the regulation of one such ligand.

## Results

### A genome-wide screen to identify novel drivers of ULBP1 expression

Many tumors and cancer cell lines express multiple NKG2D ligands, possibly due to ongoing stress responses associated with the transformed state (*Raulet et al., 2013*). To identify novel drivers of human NKG2D ligand expression in transformed cells, we employed a retroviral gene-trap mutagenesis screen using the near-haploid human cell line HAP1 (*Figure 1*) (*Carette et al., 2009*; *Carette et al., 2011*). Like many cell lines, HAP1 cells express multiple NKG2D ligands (*Figure 1—figure supplement 1*). We chose to screen for drivers of ULBP1 expression because it showed the brightest staining on HAP1 cells, making it particularly amenable to our loss-of-expression screen. Following mutagenesis, we selected for mutants with decreased expression of ULBP1 but intact expression of the unrelated GPI-anchored protein CD55 (*Figure 1A*). Selection of CD55$^+$ cells was used to reduce the fraction of selected cells that had lost ULBP1 expression due to mutations that alter cell surface expression of all proteins or of all GPI-linked proteins. In the first round of selection, we depleted ULBP1$^{high}$ cells from the mutant cell population using magnetic bead-based depletion of cells labeled with a ULBP1 antibody. After briefly expanding the selected cells, we used flow cytometry to further select for ULBP1$^{low}$CD55$^+$ cells. *Figure 1B* shows ULBP1 and CD55 expression on WT and post-selection HAP1 cells.

We employed deep-sequencing to map and quantify the frequencies of independent insertion sites of the retroviral gene-trap in selected cells and compared this with the landscape of insertions in unselected control cells. *Table 1* shows a selected 'hit list' of genes that were targeted significantly more frequently in selected ULBP1$^{low}$CD55$^+$ cells than in unselected cells. The *ULBP1* gene itself was a highly significant hit, providing a validation of this approach. Many genes encoding enzymes involved in GPI synthesis were also represented despite the selection for CD55 expression; many of these were removed from *Table 1*, for simplicity. The complete list of hits (p < 0.05) is shown in *Supplementary file 1*, along with the analysis of all independent insertions mapped in the selected data set. Raw sequencing data for the screen are available under NCBI Bioproject PRJNA284536, containing the datasets for HAP1 gene trap control cells (Accession number SAMN03703230) and cells from the ULBP1 screen (Accession number SAMN03703231). We chose hits for validation and follow-up experiments based on their statistical ranking and expectations that the corresponding proteins play roles in stress responses, protein biogenesis, or gene/mRNA regulation. The genes chosen encode ATF4 (a stress-associated transcription factor), RBM4 (an RNA-binding protein), HSPA13 (a protein chaperone), and SPCS1 and SPCS2, which are both non-catalytic subunits of the signal peptidase complex.

### Validation of hits using the CRISPR/Cas9 system and gene restoration

To confirm that selected genes from the screen regulate ULBP1, we employed the CRISPR/Cas9 mutagenesis system, targeting sites in the 5′ coding regions of each candidate gene in HAP1 cells (*Jinek et al., 2013*; *Mali et al., 2013*). The *ULBP1* gene was targeted for comparison. After HAP1 cells were transiently transfected with plasmids encoding Cas9 and guide RNAs (sgRNAs) for each candidate gene, a population of ULBP1$^{low}$ cells appeared that was absent in control transfected cells (*Figure 2—figure supplement 1*). In each case, individual ULBP1$^{low}$ cells were sorted into 96-well plates, and expanded clones were screened for mutations by PCR and sequencing. For further analysis, we selected clones with insertions or deletions that resulted in frameshift mutations in each targeted gene (*Figure 2—figure supplement 2*). Since the sites targeted were near the beginning of each coding region and the cells are haploid, the frameshift mutations are expected to result in complete loss-of-function of the corresponding proteins. Analysis of HAP1 cell lysates by Western blot confirmed the loss-of-protein expression in ATF4, RBM4, and SPCS2 mutant cell lines (data not shown).

Cells with a *ULBP1* mutation lacked cell surface ULBP1 staining altogether, as expected, whereas the other mutations analyzed resulted in a partial (twofold to threefold) decrease in cell surface expression

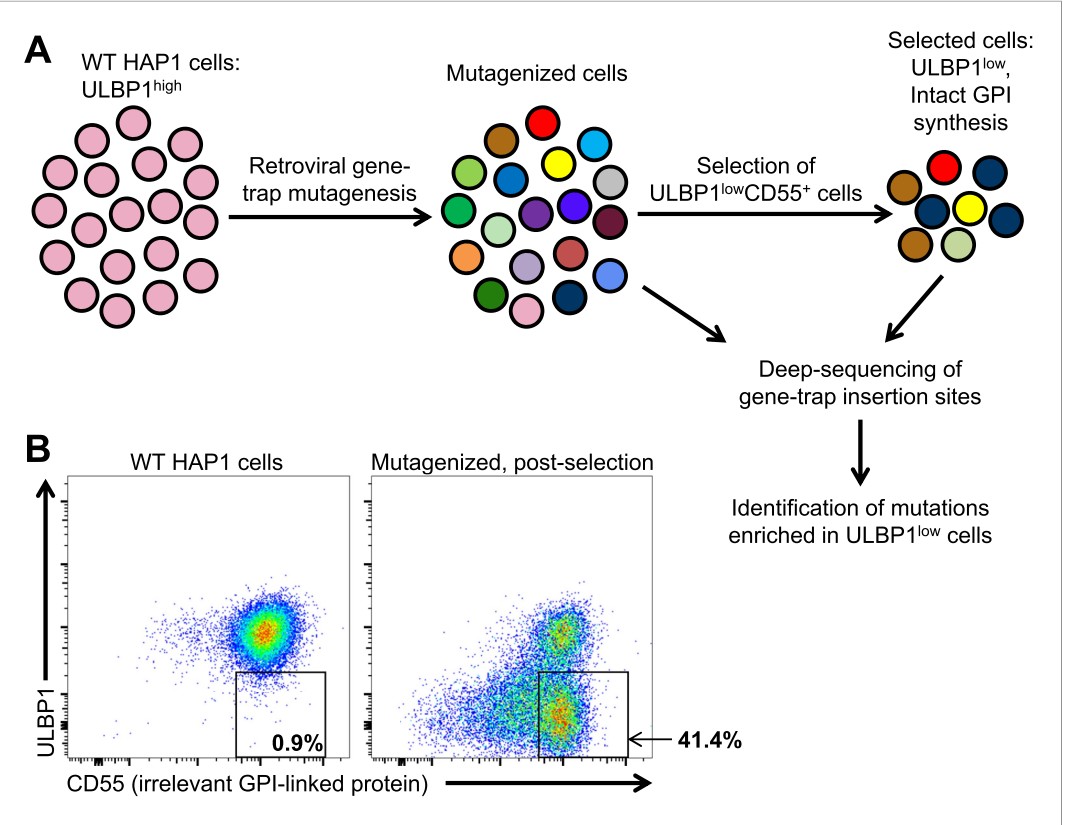

**Figure 1**. A genetic screen of a haploid human cell line to identify regulators of ULBP1 expression. (**A**) HAP1 cells (~$10^8$ cells) were transduced with a retroviral gene-trap vector. To enrich for mutant cells with decreased ULBP1 expression, we initially depleted ULBP1high cells by labeling cells with an anti-ULBP1 antibody followed by magnetic bead-based cell depletion. Following a brief recovery and expansion of the cells, we used FACS to further enrich ULBP1lowCD55+ cells. Deep-sequencing of genomic DNA from pre- and post-selection cells was used to map sites of gene-trap insertions, and mutations enriched in ULBP1low cells were identified. (**B**) Flow cytometric analysis of WT and post-selection HAP1 cells. Cells were stained for ULBP1 and CD55, an irrelevant GPI-linked protein.

The following figure supplement is available for figure 1:

**Figure supplement 1**. Expression of NKG2D ligands on HAP1 cells.

of ULBP1 (*Figure 2A*). The effect of each mutation was specific to ULBP1, as we found no change in cell surface expression of other proteins, including four other NKG2D ligands (ULBP2, ULBP3, MICA, and MICB), HLA Class I, the unrelated GPI-anchored protein CD59, or PVR and Nectin-2, the ligands for DNAM-1, another NK cell-activating receptor (*Figure 2B,C*, *Figure 2—figure supplement 3*). The minor changes in ULBP3 staining seen in *Figure 2—figure supplement 3B* were not consistently observed across experiments. The finding that the mutations each affect only ULBP1 among the NKG2D ligands tested supports the hypothesis that different NKG2D ligands are subject to distinct regulatory processes. It was surprising that *SPCS1* and *SPCS2* mutations only impacted cell surface staining of ULBP1 and not the six other membrane proteins tested, as we had expected that mutating components of the signal peptidase complex would cause a more generalized defect in cell surface protein expression (see 'Discussion'). In all cases, ULBP1 expression on mutant lines could be restored by re-expressing the gene of interest with a doxycycline-inducible lentiviral vector (*Figure 2D*). These findings established that ATF4, RBM4, HSPA13, SPCS1, and SPCS2 each contribute partially to cell surface display of ULBP1 in HAP1 cells in steady-state culture conditions.

Consistent with their presumptive roles in gene expression and/or splicing, *ATF4* and *RBM4* KO cells each showed decreased amounts of canonically spliced *ULBP1* mRNA commensurate with the change in cell surface expression (*Figure 2E*). Consistent with their roles as chaperones and

Table 1. Selected list of genes enriched for gene-trap insertions after selection of ULBP1^low CD55^+ cells

| Gene symbol | Function/Process | p-value |
| --- | --- | --- |
| PIGW | GPI synthesis/anchoring | 1.12E-196 |
| PIGQ | GPI synthesis/anchoring | 3.26E-155 |
| PIGB | GPI synthesis/anchoring | 2.38E-103 |
| **ULBP1** | **NKG2D ligand** | **2.65E-76** |
| PIGO | GPI synthesis/anchoring | 1.49E-65 |
| **RBM4** | **RNA-binding protein** | **1.29E-24** |
| PIGV | GPI synthesis/anchoring | 4.43E-23 |
| **SPCS1** | **Non-catalytic subunit of signal peptidase complex** | **1.25E-15** |
| PIGM | GPI synthesis/anchoring | 6.06E-14 |
| C1GALT1C1 | Protein O-linked glycosylation | 2.22E-13 |
| SLC35A1 | Golgi-localized CMP-sialic acid transporter | 1.77E-12 |
| ST3GAL2 | Sialyltransferase | 1.64E-11 |
| **SPCS2** | **Non-catalytic subunit of signal peptidase complex** | **6.20E-09** |
| **HSPA13** | **Microsome-associated protein with ATPase activity** | **1.24E-05** |
| FLJ37453 | Non-coding RNA | 0.00115 |
| SLC17A9* | Vesicular nucleotide transporter | 0.00715 |
| RPS25 | Ribosomal protein | 0.00715 |
| **ATF4** | **Stress-induced transcription factor** | **0.0206** |
| PMM2 | Oligosaccharide synthesis, protein glycosylation | 0.0234 |
| NCRNA00167 | Non-coding RNA | 0.0363 |
| CRNKL1* | Pre-mRNA splicing | 0.0363 |
| ICK | Intestinal cell kinase, MAPK-related | 0.0363 |
| TBC1D19 | TBC domain-containing protein | 0.0416 |
| ZNF236 | Zinc-finger protein | 0.0496 |

The gene symbols of hits ($p < 0.05$) are shown with a brief description of known or predicted gene functions. A p-value of enrichment was determined using Fisher's exact test, followed by correction for the false discovery rate. The list was manually curated to remove known genes that have occurred in several unrelated screens using the same cells, perhaps indicating pleiotropic effects. For simplicity, a number of genes related to GPI biosynthesis and anchoring were removed. Bold text indicates genes confirmed in this study to impact ULBP1 expression. Blue text indicates genes involved in GPI biosynthesis and anchoring. Red text indicates genes involved in protein glycosylation. Asterisks indicate two genes (*SLC17A9* and *CRNKL1*) that, when targeted with CRISPR/Cas9, failed to result in decreased ULBP1 expression.

protein-processing components, *HSPA13*, *SPCS1*, and *SPCS2* KO cells all showed WT amounts of *ULBP1* mRNA.

## Stepwise regulation of ULBP1 expression

To assess whether the genes implicated in ULBP1 expression work independently or in common pathways, we generated and analyzed double and triple-mutant cell lines. We generated *ATF4/RBM4* double-mutant cells by targeting *ATF4* in the *RBM4* KO line, and triple-mutant cells by further targeting either *HSPA13* or *SPCS2* in the double-mutant line. Notably, we observed stepwise decreases in ULBP1 expression with each additional mutation, with triple-mutant cells showing up to a 20-fold reduction in ULBP1 expression compared to WT cells (*Figure 3A*). These data suggested that the genes tested (with the likely exception of SPCS1 vs SPCS2) contribute largely independently to

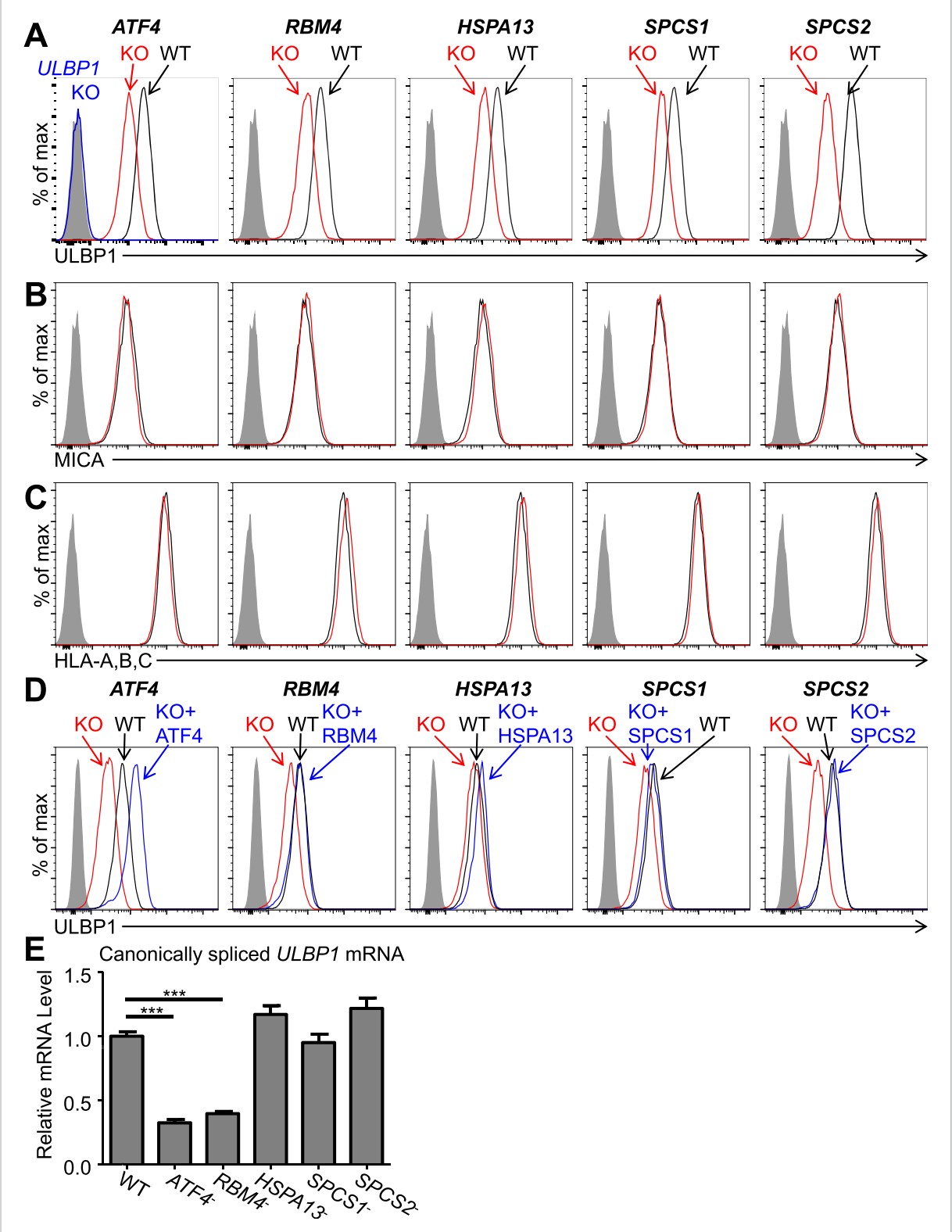

**Figure 2**. Decreased ULBP1 expression upon targeted mutation of screen hits. (**A–C**) Flow cytometric analysis of cell surface expression of ULBP1 (**A**), the NKG2D ligand MICA (**B**), or pan-HLA Class I (**C**) on WT and mutant HAP1 cells. WT and mutant (KO) cells are represented as black and red histograms, respectively. The shaded gray histogram represents isotype control staining. The blue trace in panel **A** shows staining of *ULBP1* KO HAP1 cells and matches isotype control staining. Data are representative of at least three independent experiments. (**D**) To restore expression of ULBP1 drivers, mutant

*Figure 2. Continued*

cell lines were transduced with a doxycycline-inducible lentiviral vector containing the gene of interest. Cells were treated for 24 hr with doxycycline (Dox) at a final concentration of 100 ng/ml for ATF4 and 1000 ng/ml for all other genes. After treatment, cells were analyzed by flow cytometry. Black histograms: WT cells transduced with control vector, +Dox. Red histograms: mutant cells transduced with Dox-inducible gene of interest, −Dox. Blue histograms: mutant cells transduced with Dox-inducible gene of interest, +Dox. The shaded gray histogram represents isotype control staining. Data are representative of three independent experiments. (E) RT-qPCR analysis of canonically spliced *ULBP1* mRNA expression levels in WT and mutant HAP1 cells. Expression levels were normalized to *ACTB*, *GAPDH*, and *HPRT1* and are shown as mean ±SE. The data were analyzed by 1-way ANOVA with Dunnet's multiple comparisons test comparing all samples to WT. ***$p < 0.001$. Data are representative of three independent experiments. In one out of three total experiments performed, the level of *ULBP1* mRNA was significantly increased in the *SPCS2* mutant compared to WT.

The following figure supplements are available for figure 2:

**Figure supplement 1**. Mutagenesis of screen 'hits' with the CRISPR/Cas9 system.

**Figure supplement 2**. Sequences of CRISPR/Cas9 target sites in HAP1 cells.

**Figure supplement 3**. Expression of additional NK cell ligands is unchanged on mutant HAP1 cells.

steady-state ULBP1 expression in HAP1 cells and support a model in which constitutive NKG2D ligand expression in cell lines is due, at least in some instances, to the contribution of several pathways that act cumulatively. *ATF4/RBM4* double mutants showed a larger decrease in canonically spliced *ULBP1* mRNA than either single mutant, suggesting that ATF4 and RBM4 may act independently in regulating the amounts of *ULBP1* mRNA (*Figure 3B*). There were no greater decreases in *ULBP1* mRNAs when mutations in either *HSPA13* or *SPCS2* were added to the double-mutant cells, as expected if they act co-translationally or post-translationally.

## ATF4 supports ULBP1 expression on multiple tumor-derived cell lines

Having validated several hits from our screen, we investigated the roles of ATF4 and RBM4 in regulating ULBP1 expression. To address whether ATF4 regulates *ULBP1* in other cell types, we mutated *ATF4* in the K-562 chronic myelogenous leukemia cell line and the Jurkat acute T-cell leukemia cell line. *ATF4* KO K-562 and Jurkat cells were generated using Cas9 and a pair of sgRNAs that flank the *ATF4* gene. The entire *ATF4* gene was deleted in the resulting mutant lines, eliminating the possibility of the mutant cells expressing a functional ATF4 protein fragment. HAP1 cells carrying a similar deletion of *ATF4* had an identical phenotype to the *ATF4* frameshift mutant line described above (data not shown). WT K-562 cells had relatively low ULBP1 expression, and mutation of *ATF4* resulted in the complete disappearance of ULBP1 from the cell surface, as well as an 11-fold reduction in *ULBP1* mRNA (*Figure 4A,B*). In contrast, ATF4-deficiency in Jurkat cells resulted in a modest reduction in *ULBP1* mRNA and a barely detectable reduction in ULBP1 cell surface staining. By comparison, HAP1 cells showed intermediate effects of ATF4-deficiency. The results suggest that ATF4 drives basal *ULBP1* expression in multiple tumor cell lines, perhaps reflecting constitutive activation of underlying stress pathways.

## ATF4 drives *ULBP1* transcription in response to cell stress

ATF4 is induced in a variety of stress conditions that arise in unhealthy, infected, and transformed cells; coupling these stress conditions to NKG2D ligand expression may enable destruction of undesirable cells. Such stresses, including amino acid starvation, the unfolded protein response (UPR), oxidative stress, and the presence of dsRNA, induce ATF4 through phosphorylation of the translation initiation factor eIF2α (*Rutkowski and Kaufman, 2003*; *Suragani et al., 2012*). Phosphorylation of eIF2α globally suppresses protein translation but selectively activates the translation of ATF4 (*Harding et al., 2000*). The transcriptional targets of ATF4 include amino acid transporters and protein chaperones, which in combination with the overall reduction in protein synthesis help mitigate the cellular stress (*Harding et al., 2003*). However, ATF4 expression also drives the expression of the pro-apoptotic transcription factor CHOP (GADD153), suggesting that ATF4 not only drives adaptation to stress but also promotes cell death if the stress cannot be overcome (*Tabas and Ron, 2011*;

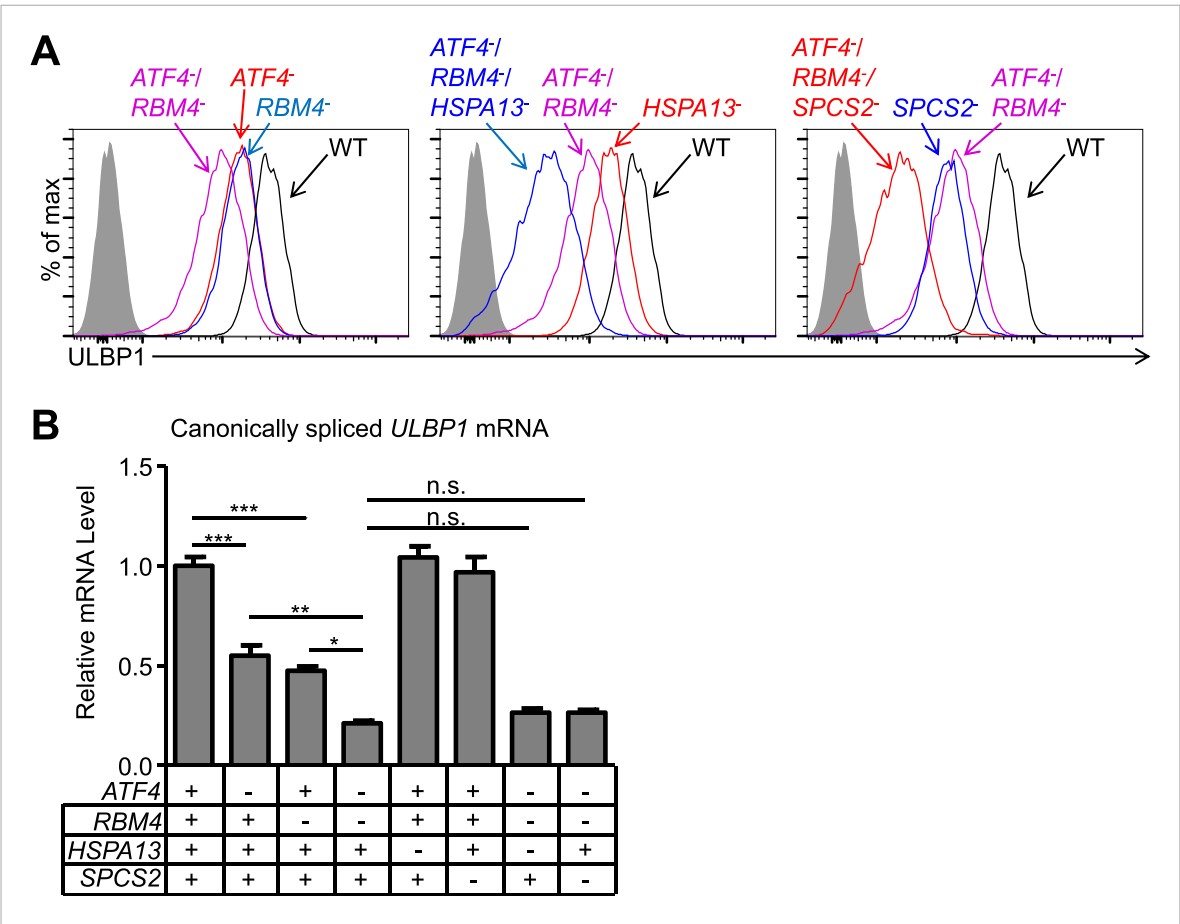

**Figure 3**. Double and triple-mutant cell lines show stepwise decreases in ULBP1 expression. (**A**) Flow cytometric analysis of ULBP1 expression on single, double, and triple-mutant HAP1 cells. *ATF4⁻/RBM4⁻* double-mutant cells were generated by mutagenesis of *ATF4* in *RBM4⁻* cells. Triple-mutant cells were generated by mutagenesis of *HSPA13* or *SPCS2* in *ATF4⁻/RBM4⁻* double-mutant cells. Shaded gray histograms represent isotype control staining. Data are representative of three independent experiments. (**B**) RT-qPCR analysis of canonically spliced *ULBP1* mRNA expression levels in the cells described in (**A**). Expression levels were normalized to *ACTB*, *GAPDH*, and *HPRT1* and are shown as mean ±SE. The data were analyzed by 1-way ANOVA with Bonferroni's multiple comparisons test and substantive significant differences are shown. Data are representative of three independent experiments. *$p < 0.05$, **$p < 0.01$, ***$p < 0.001$, n.s.: not significant.

*Han et al., 2013*). Increased ATF4 expression has been detected in some tumors (*Bi et al., 2005*), providing a possible mechanism for coupling malignant transformation to expression of ULBP1, and consequently to tumor immunosurveillance.

To impart stress known to induce ATF4 expression, we treated cells with the drug histidinol (HisOH), which competitively inhibits histidine–tRNA charging, thus mimicking amino acid starvation (*Hansen et al., 1972*; *Thiaville et al., 2008*), or thapsigargin (Tg), which induces the UPR (*Oslowski and Urano, 2011*). Histidinol or thapsigargin treatment each significantly induced *ULBP1* mRNA in WT K-562, HAP1, and Jurkat cells, but induction was abrogated or severely blunted when cells lacked ATF4 (*Figure 5A*). Induction of *ULBP1* transcription by the proteasome inhibitor MG132 (*Butler et al., 2009*) was not prevented in the *ATF4*-mutant cells (data not shown), indicating that certain cellular perturbations transcriptionally activate *ULBP1* independently of ATF4. RNA-Seq analysis of HAP1 cells confirmed ATF4-dependent upregulation of *ULBP1* mRNA in response to histidinol treatment (*Figure 5—figure supplement 1*). RT-qPCR analysis of other NKG2D ligands expressed by HAP1 cells showed that histidinol treatment caused an increase in *ULBP2* mRNA that was partially ATF4-dependent, while *MICA* expression was slightly decreased in an ATF4-independent manner (*Figure 5—figure supplement 2*). *ULBP3* and *MICB* transcripts were not affected by histidinol treatment or loss of ATF4 (*Figure 5—figure supplement 2*). *ULBP4*, *ULBP5*, and *ULBP6* mRNAs were absent or barely detectable by RNA-Seq

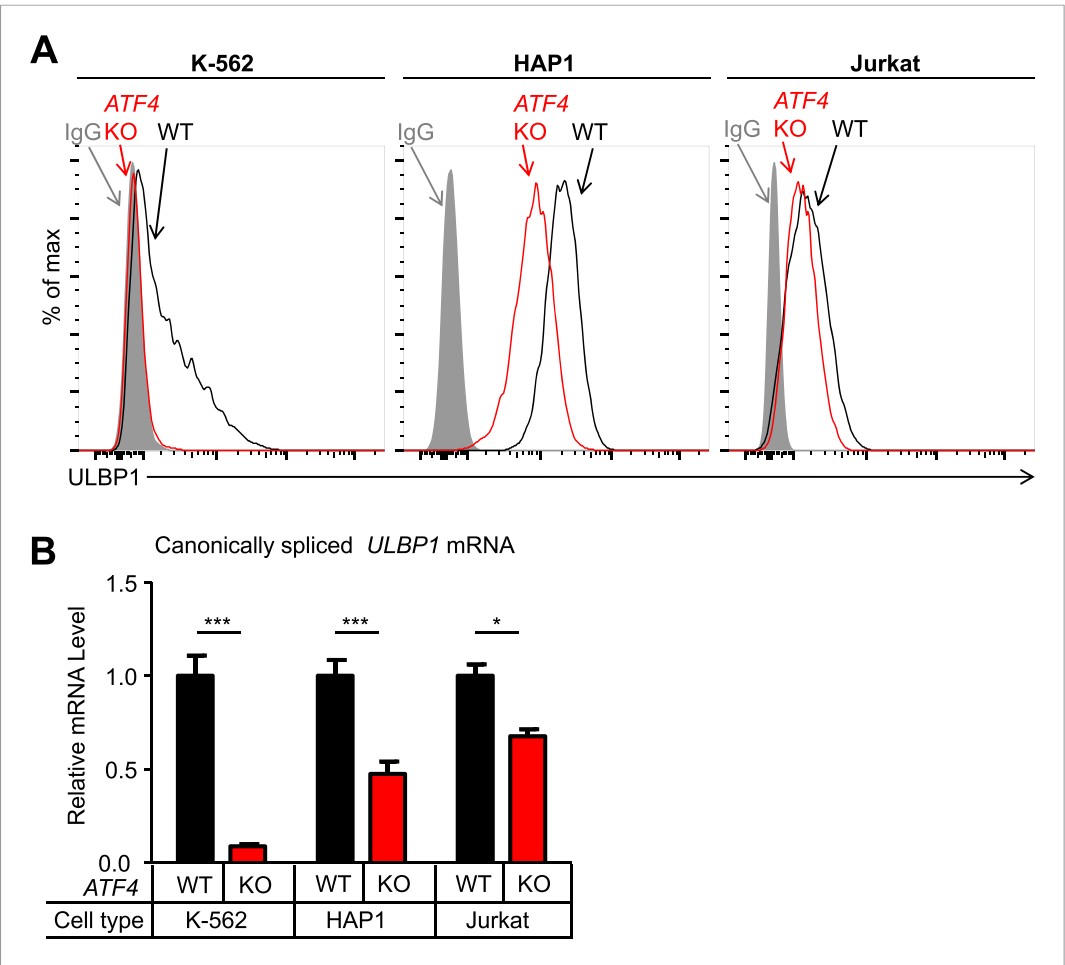

**Figure 4**. ATF4 drives basal ULBP1 expression in multiple cell lines. (**A**) Flow cytometric analysis of ULBP1 expression on WT versus *ATF4* KO variants of K-562, HAP1, and Jurkat cells. Data are representative of three independent experiments. (**B**) RT-qPCR analysis of canonically spliced *ULBP1* mRNA expression levels in the cells described in (**A**). Expression levels were normalized to *ACTB*, *GAPDH*, and *HPRT1* and are shown as mean ±SE. Expression in WT cells was set to '1.0' for each cell type; the different cell types are not comparable in this experiment. The data were analyzed by 2-way ANOVA with Bonferroni's multiple comparisons test. Data are representative of three independent experiments, though one of the three Jurkat analyses did not show a significant difference. *p < 0.05, ***p < 0.001.

analysis of untreated HAP1 cells, and those genes showed no sign of induction by histidinol (data not shown). These data suggest that the ATF4-mediated stress response induces expression of *ULBP1*, and to a minor extent *ULBP2*, but not most other NKG2D ligands.

Stress-induced cell surface ULBP1 protein expression was also examined. Although histidinol and thapsigargin induce transcription of ATF4-regulated genes, they also inhibit protein synthesis, both directly (e.g., by limiting the availability of charged histidine tRNAs) and indirectly, by causing eIF2α phosphorylation. These opposing effects make it difficult to predict whether stress induction will result in increased amounts of any specific protein encoded by an ATF4-regulated gene. Nevertheless, in K-562 cells, we observed a dramatic ATF4-dependent induction of cell surface ULBP1 by histidinol (*Figure 5B,D*). In HAP1 and Jurkat cells, in contrast, addition of histidinol induced ULBP1 cell surface expression slightly in WT cells, but caused decreased ULBP1 expression in *ATF4*-mutant cells (*Figure 5B,D*), consistent with the aforementioned inhibition of protein synthesis resulting from histidinol treatments. Indeed, histidinol treatments resulted in substantially decreased cell surface expression of HLA proteins, which are not regulated by ATF4 (*Figure 5C,E*). As summarized in *Figure 5D*, the overall effect of ATF4 induction by histidinol was to induce and/or

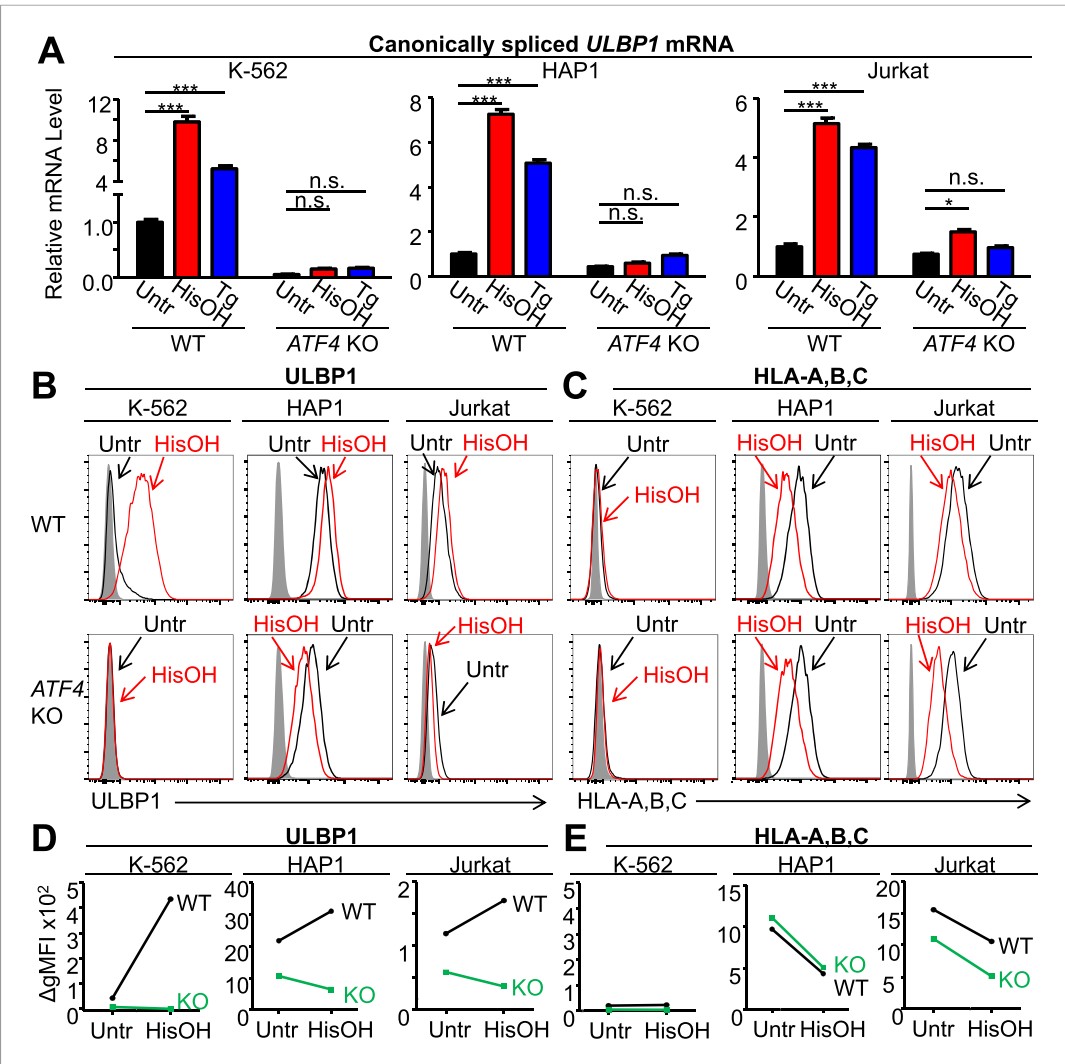

**Figure 5**. ATF4 drives increased expression of ULBP1 mRNA and surface protein in response to cell stress. (**A**) Cells were treated for 24 hr with 2 mM histidinol (HisOH) to mimic amino acid starvation or 300 nM thapsigargin (Tg) to induce the unfolded protein response. RNA was isolated from treated and control cells, and canonically spliced *ULBP1* mRNA levels were determined by RT-qPCR. Expression levels were normalized to *ACTB*, *GAPDH*, and *HPRT1* and are shown as mean ±SE. Expression in untreated WT cells was set to '1.0' for each cell type; the different cell types are not comparable in this experiment. For reference, the Cq values for *ULBP1* in untreated WT cells were 32.2 for K-562 cells, 27.9 for HAP1 cells, and 30.5 for Jurkat cells. The data were analyzed by 2-way ANOVA with Bonferroni's multiple comparisons test and are representative of three independent experiments, though in one of the three analyses of *ATF4* KO Jurkat cells, histidinol-treated cells trended higher than untreated cells but did not reach significance. *$p < 0.05$, ***$p < 0.001$, n.s.: not significant. (**B**, **C**) Flow cytometric analysis of ULBP1 (**B**) and HLA Class I expression (**C**) on cells treated with histidinol as in (**A**). (**D**, **E**) Quantification of surface staining shown in (**B**) and (**C**). Data are plotted as the geometric mean fluorescence intensity of the specific stain minus the intensity of the isotype control (ΔgMFI). Data are representative of three independent experiments.
The following figure supplements are available for figure 5:

**Figure supplement 1**. Analysis of *ULBP1* expression by RNA-Seq.

**Figure supplement 2**. Analysis of other NKG2D ligands in response to cell stress.

maintain ULBP1 cell surface expression in the face of global protein synthesis inhibition associated with this stress pathway.

To address whether ATF4 directly regulates *ULBP1* transcription, we used ChIP-Seq analysis to determine whether it binds to *ULBP1* regulatory elements in histidinol-treated HAP1 cells. ChIP-Seq with three independent ATF4 antibodies showed a strong peak of ATF4 binding associated with the *ULBP1* promoter, along with a smaller peak ~27 kb downstream in the intergenic region, which could function as an enhancer. No other notable ATF4 binding was observed in the 266 kb interval shown, which includes genes encoding five other functional NKG2D ligands (*ULBP2-6; RAET1K* is a pseudogene), at least two of which are expressed in HAP1 cells (*ULBP2* and *ULBP3*) (*Figure 6A*, data not shown). The *MICA* and *MICB* genes were also devoid of ATF4 binding (data not shown). HOMER Motif Analysis software identified ATF4-binding motifs in both peaks (*Figure 6B,C*) (*Heinz et al., 2010*). Additional ATF4-binding motifs were present in the segment shown but were not bound by ATF4. ATF4 binding at the *ULBP1* promoter was confirmed by conventional ChIP-qPCR, which also demonstrated that ATF4 binds to the *ULBP1* locus in HAP1 cells under steady-state conditions (no histidinol), and that ATF4 binding is sharply enhanced after histidinol treatment (*Figure 6D,E*). A similar pattern of ATF4 binding was observed at the *ASNS* promoter, a known target of ATF4, but not at a negative control site in the *ASNS* gene body (*Figure 6D,E*, *Figure 6—figure supplement 1*) (*Chen et al., 2004*).

We used luciferase reporter assays to confirm direct activation of the *ULBP1* promoter by ATF4. We inserted previously described fragments of the *ULBP1* promoter (*López-Soto et al., 2006*) and 5' UTR upstream of a luciferase reporter and tested promoter responsiveness to ATF4 overexpression in *ATF4*-deficient HAP1 cells (*Figure 6—figure supplement 2*). ATF4 over-expression increased *ULBP1* promoter activity, and the fold-induction was similar for the longer and shorter promoter fragments tested. To identify the relevant ATF4-binding site(s), we employed the shorter promoter fragment and mutated either the ATF4 motif we had identified, or each of two Cyclic AMP Response Element (CRE) sites, which were previously shown to drive *ULBP1* promoter activity and are similar to the ATF4 consensus sequence. Mutation of the ATF4 motif 138–147 bp upstream of the transcription start site completely ablated promoter activation by ATF4 (see mutant 1, m1), while mutation of the CRE sites did not reduce induction by ATF4. We conclude that ATF4 directly transactivates the ULBP1 promoter, and that a single ATF4-binding site drives the bulk of this response.

## RBM4 suppresses alternative splicing of *ULBP1* mRNA

RBM4 is an RNA-binding protein that has been implicated in several steps of gene regulation but has been most studied for its regulation of RNA splicing (*Markus and Morris, 2009*). We compared WT and *RBM4* KO HAP1 cells by RNA-Seq to determine a potential role of RBM4 in splicing of *ULBP1* transcripts. As a result, we discovered a novel isoform of *ULBP1* that to our knowledge has not been previously reported (*Figure 7A*). The sequence of the novel isoform has been deposited in GenBank (Accession number KT591165). Analysis of sequence reads that aligned to splice junctions revealed that most transcripts in WT cells exhibited the expected splicing pattern to encode the normal ULBP1 protein (*Figure 7A,B*). In *RBM4* KO cells, in contrast, approximately 60% of transcripts had undergone an alternative splicing event of the first and second exons in which the 5' splice site at the 3' border of the first exon was ignored in favor of a 5' splice site located 1.3 kb downstream within the first intron (*Figure 7A*). Splicing of the downstream exons of *ULBP1* appeared normal in *RBM4* KO cells. The alternatively spliced transcript in *RBM4* KO cells encodes a premature stop codon shortly after the predicted ER signal peptide and is thus unlikely to encode a functional protein fragment (*Figure 7C*). We did not observe differential splicing of the other NKG2D ligands expressed by HAP1 cells (*MICA, MICB, ULBP2,* and *ULBP3, Figure 7—figure supplement 1*).

The alternative splicing of *ULBP1* transcripts in *RBM4* KO cells was confirmed by RT-qPCR analysis (*Figure 7C–E, Figure 7—figure supplement 2*). Canonically spliced *ULBP1* transcripts were less abundant in *RBM4* KO cells, while the alternatively spliced transcripts were more abundant; the overall abundance of *ULBP1* transcripts was unchanged as a result of RBM4-deficiency. The reduction in canonically spliced transcripts was commensurate with the reduction in cell surface ULBP1 protein, suggesting that RBM4 impacts cell surface expression of *ULBP1* primarily by facilitating the correct splicing of the first two exons of *ULBP1* transcripts.

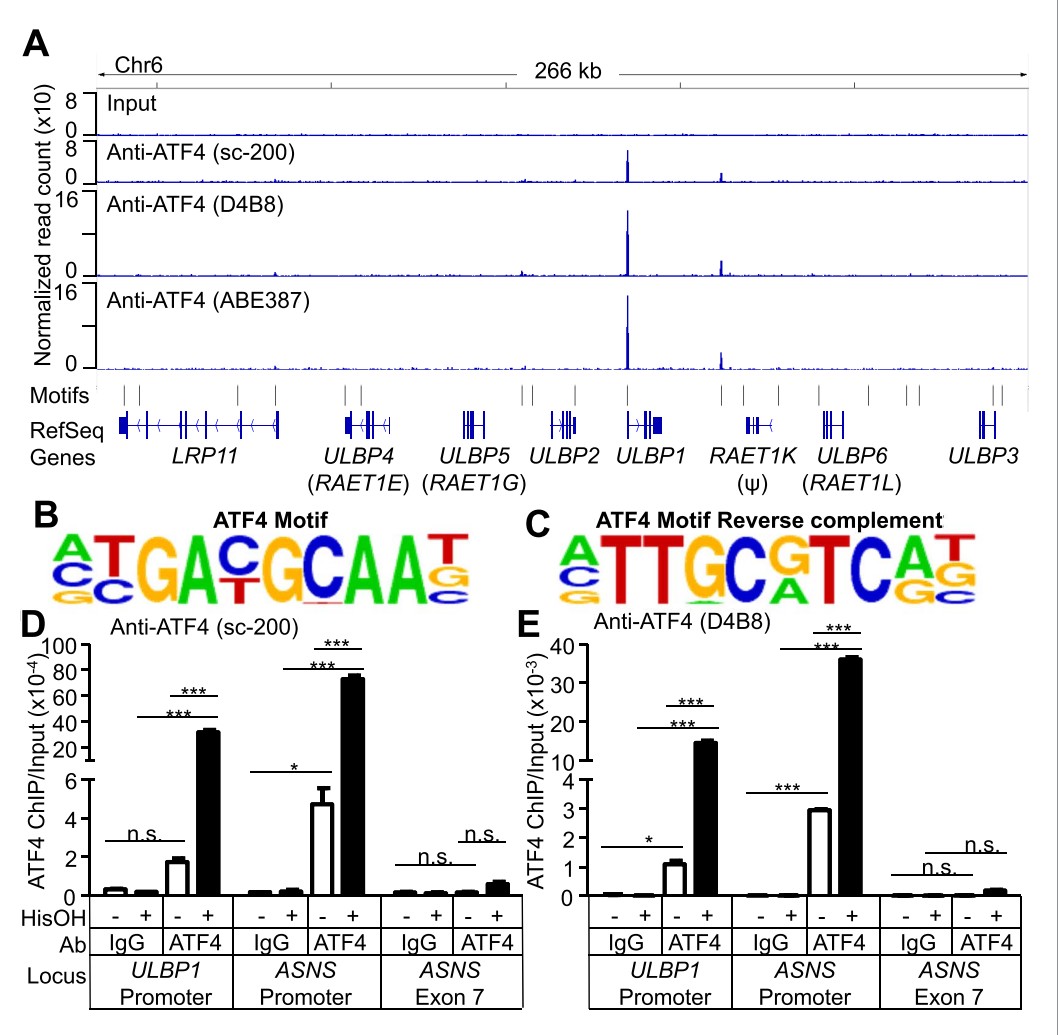

**Figure 6**. ATF4 is bound to the ULBP1 promoter and a potential downstream regulatory element. (**A**) ATF4 ChIP-Seq: HAP1 cells were treated with 2 mM histidinol (HisOH) for 24 hr, followed by formaldehyde cross-linking. ATF4-bound chromatin was immunoprecipitated using three independent anti-ATF4 antibodies, and the isolated DNA was sequenced and aligned to the human genome (hg19). Locations of consensus ATF4-binding motifs are indicated. *RAET1K* is a pseudogene (ψ). (**B**, **C**) HOMER Motif Analysis software identified the ATF4 binding motif (**B**) and its reverse-complement (**C**). (**D**, **E**) Conventional ChIP-qPCR of ATF4 using the antibody sc-200 (**D**) or D4B8 (**E**). Samples were treated as in (**A**), followed by qPCR. ChIP signal was normalized to the amount of Input DNA for each sample. Data are plotted as mean ±SE and were analyzed by 2-way ANOVA with Bonferroni's multiple comparisons test. Representative data are shown. The ChIP-qPCR experiment was performed twice using both untreated and HisOH-treated cells, and a third time using only HisOH-treated cells. *$p < 0.05$, ***$p < 0.001$, n.s.: not significant. Raw sequencing data corresponding to ATF4 ChIP-Seq have been deposited to GEO. Accession: GSE69304.

The following figure supplements are available for figure 6:

**Figure supplement 1**. ATF4 ChIP-Seq signal at the *ASNS* promoter, a known ATF4-binding site.

**Figure supplement 2**. ATF4 directly activates the *ULBP1* promoter.

## Discussion

The mutant screen identified several regulators of ULBP1 in HAP1 cells. Loss-of-function mutations in each of the corresponding genes resulted in relatively modest reductions in cell surface ULBP1 when examined separately. The capacity of the screen to detect mutations that each have relatively small

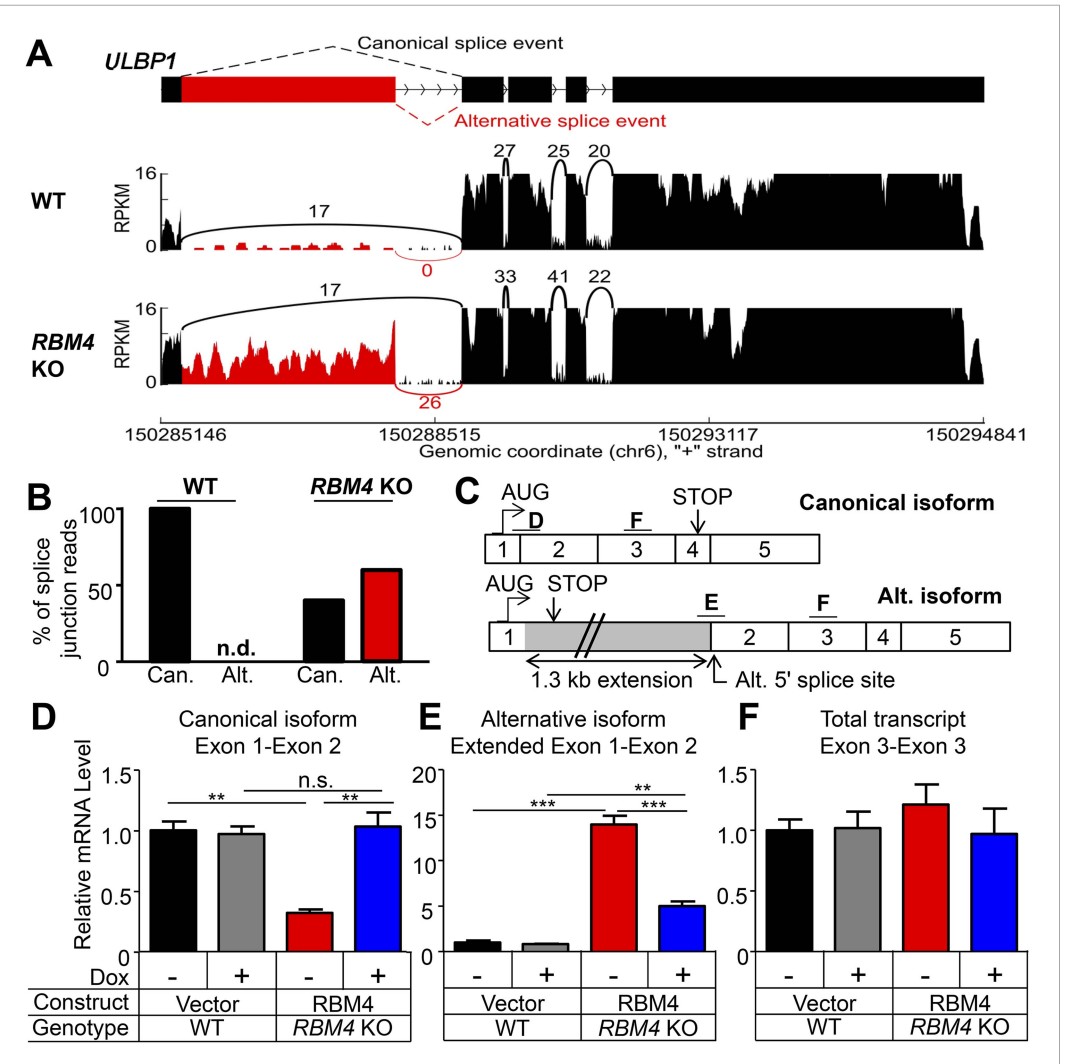

**Figure 7**. RBM4 suppresses the alternative splicing of *ULBP1* mRNA. (**A**) 'Sashimi plots' of RNA-Seq reads aligning to *ULBP1* mRNA from WT and *RBM4* KO HAP1 cells. Peak height represents the number of reads aligning to a given segment (RPKM). Arcs connect splice junctions identified by reads spanning the splice junction; the number of reads aligning to each splice junction is indicated. (**B**) Quantification of splice junction-spanning reads mapping to canonical or alternatively spliced *ULBP1* mRNA. Reads spanning the canonical or alternative splice junction are expressed as a percent of the total junction-spanning reads (canonical + alternative). n.d.: not detected. (**C**) Diagram of the canonical and alternatively spliced isoforms of the *ULBP1* mRNA. The alternative isoform contains a premature stop codon and is unlikely to produce a functional protein product. The PCR amplicons used in (**D–F**) are indicated. (**D–F**) RT-qPCR analysis of *ULBP1* transcript levels in WT and *RBM4* KO HAP1 cells. Cells were transduced with a doxycycline-inducible RBM4 vector or control vector and treated with 100 ng/ml Dox for 24 hr. The data were analyzed by 1-way ANOVA with Bonferroni's multiple comparisons test and are representative of three independent experiments. **p < 0.01, ***p < 0.001, n.s.: not significant.

The following figure supplements are available for figure 7:

**Figure supplement 1**. Splicing of other NKG2D ligands is unchanged in *RBM4* KO HAP1 cells.

**Figure supplement 2**. Additional analysis of *ULBP1* transcript in WT and *RBM4* KO HAP1 cells.

effects demonstrates its remarkable power to identify pathways that function in a partially redundant fashion. Overall, the data suggest that most of the pathways studied control different steps in the biogenesis of ULBP1, with ATF4 controlling transcription, RBM4 regulating RNA splicing, and

HSPA13, SCPS1, and SPCS2 regulating co-translational or post-translational steps such as protein processing. The power of the CRISPR/Cas9 method to rapidly generate double- and triple-mutant cell lines led to the finding that disabling multiple regulators resulted in much more striking reductions in ULBP1 expression than disabling them individually. Thus, each regulator contributes incrementally to ULBP1 expression. This finding lends support to the hypothesis that stress pathways might cumulatively control the amount of a given NKG2D ligand displayed on the cell to define a quantitative 'threat level' confronting a cell (*Raulet et al., 2013*). Previously published evidence demonstrated that greater NK-dependent tumor elimination occurs with increasing amounts of cell surface NKG2D ligand (*Diefenbach et al., 2001*), consistent with the 'threat-level' hypothesis.

When characterizing the phenotypes of our mutant HAP1 cell lines, we were surprised to find that in each case, ULBP1 was the only NKG2D ligand with substantially altered expression, the one exception being an increase in MICB expression observed on *HSPA13* KO cells. These data provide strong support for the proposal that different ligands are specialized with respect to their regulatory mechanisms, providing the capacity to respond to specific classes of threats (*Raulet et al., 2013*). In contrast, certain other regulatory pathways, such as the DNA damage response, regulate several NKG2D ligands similarly (*Gasser et al., 2005*; *Soriani et al., 2009*; *Raulet et al., 2013*). The system as a whole, therefore, encompasses both generic and specific regulators.

## ATF4 and transcription of ULBP1

A key finding was that ATF4, a mediator of several stress responses, transcriptionally activates *ULBP1*. The data show that ATF4 binds to two locations at the *ULBP1* locus, including the promoter region, that an ATF4 site in the *ULBP1* promoter is essential for ATF4-induced transcription, and that ATF4 drives *ULBP1* transcription in wild-type cells. No additional ATF4 binding or transcriptional induction was detected for most other NKG2D ligand genes, although *ULBP2* was modestly upregulated by histidinol treatment in a manner partially dependent on ATF4. Since ATF4 was not bound to the *ULBP2* promoter, we speculate that *ULBP2* transcription is regulated by a transcription factor that is induced by ATF4, or that ATF4 binding to the sites in the *ULBP1* gene modestly activates the neighboring *ULBP2* gene.

Disruption of *ATF4* resulted in reduced ULBP1 expression under steady-state conditions in all three of the cell lines tested. These data suggest that ATF4 activity caused by constitutive stress in tumor cell lines contributes to ULBP1 expression. This finding has physiological relevance, as overexpression of UPR components has been found in primary tumors and cell lines derived from breast cancers (*Fernandez et al., 2000*; *Fujimoto et al., 2003*), hepatocellular carcinomas (*Shuda et al., 2003*), and multiple myeloma (*Reimold et al., 1996*; *Munshi et al., 2004*).

The three cell lines we examined illustrate the different ways that ATF4 can support steady-state ULBP1 expression in different contexts. In some cases, expression of ULBP1 may be critically dependent on ATF4. This is the case in unstimulated K-562 cells in which basal ULBP1 expression was detected (though weakly) in WT cells but was absent from the surface of *ATF4* KO cells. In other cases, exemplified by HAP1 and Jurkat cells, numerous factors contribute to ULBP1 expression, with ATF4 being only one of them. Those cells had higher basal ULBP1 expression than K-562 cells and showed a modest decrease in ULBP1 expression when *ATF4* was mutated.

The impact of ATF4 on *ULBP1* transcription was most notable when additional stressors were applied that induce ATF4 expression. Outcomes at the level of protein expression are difficult to predict a priori, because these stressors induce transcription of ATF4 target genes, but also cause global suppression of translation as a result of eIF2α phosphorylation. We observed a range of outcomes in different cell lines. Amino acid starvation of K-562 cells, imparted by histidinol treatment, caused a substantial increase in *ULBP1* mRNA and surface protein, and the response was ablated in *ATF4* KO cells. In this case, the increase in *ULBP1* transcription overwhelmed the global decrease in translation. In contrast, in HAP1 and Jurkat cells, amino acid starvation of WT cells caused a fivefold to sevenfold increase in *ULBP1* mRNA, but a much smaller increase in ULBP1 protein at the cell surface. Nevertheless, the impact of ATF4 was made apparent by the observation that ULBP1 protein levels *decreased* on the surface of similarly stressed *ATF4* KO cells. In these cell lines, therefore, the induction of *ULBP1* transcription by ATF4 serves to *maintain* (and slightly increase) expression of ULBP1 protein in the face of cellular stress. Maintaining ULBP1 expression that is induced by distinct regulatory pathways is important in cells where translational inhibition is likely to occur, such as in hypoxic tumors.

Our finding that *ULBP1* expression is regulated by ATF4 fits the existing paradigm for ATF4 target genes. ATF4 is induced by amino acid starvation, oxidative stress, hypoxia, and the UPR. ATF4 target genes include amino acid transporters, anti-oxidant response genes, angiogenic factors, and protein chaperones, and thus, ATF4 initially promotes cell survival against these insults (*Jain, 2005*; *Roybal et al., 2005*; *Han et al., 2013*). Previous studies have shown that ATF4 expression is frequently induced in primary tumors and tumor cell lines (*Fernandez et al., 2000*; *Shuda et al., 2003*; *Ma and Hendershot, 2004*). Inadequate vascularization can cause tumors to become hypoxic and nutrient-starved, two conditions known to induce ATF4. Indeed, ATF4 expression is highest in hypoxic regions of tumors (*Ameri et al., 2004*; *Bi et al., 2005*). However, the pro-apoptotic targets of ATF4, such as *CHOP*, provide an important safety mechanism for the host (*Tabas and Ron, 2011*). Without a pro-apoptotic component, the transcriptional targets of ATF4 would be ripe for exploitation by expanding tumors, which are often poorly vascularized, and consequently both hypoxic and starved. Upregulation of ULBP1 by ATF4 provides an additional layer of host defense, marking potentially dangerous cells for elimination by the immune system. Although not explored in this study, induction of ULBP1 by ATF4 may also play a role in antiviral immunity. As mentioned above, ATF4 expression is induced by phosphorylation of eIF2α. Viral infection can activate PERK and PKR, which phosphorylate eIF2α and therefore induce ATF4 expression (*Mohr and Sonenberg, 2012*; *Jheng et al., 2014*).

## RBM4 and splicing of *ULBP1* transcripts

We identified RBM4 as a second factor responsible for steady-state ULBP1 expression in HAP1 cells. RBM4-deficient cells exhibited a twofold to threefold reduction in cell surface ULBP1 and the corresponding canonical *ULBP1* mRNA isoform. Interestingly, RBM4 suppresses a novel alternative splicing event in the *ULBP1* transcript. In the absence of RBM4, total *ULBP1* mRNA levels were unchanged, but the alternatively spliced isoform was upregulated while the canonically spliced isoform was downregulated. The alternative *ULBP1* transcript contains a premature stop codon early in the protein-coding sequence and is unlikely to produce a functional protein. Transcripts containing premature stop codons can be targets for nonsense-mediated decay (NMD) (*Maquat, 2004*), but we could nonetheless detect the alternative *ULBP1* transcript by RNA-Seq. The degree to which the alternative *ULBP1* transcript is degraded by NMD is presently unclear. RBM4 has also been implicated in other forms of gene regulation, including translation and microRNA-mediated gene silencing (*Lin and Tarn, 2009*; *Uniacke et al., 2012*). In HAP1 cells, however, the change in cell surface ULBP1 in *RBM4* KO cells was commensurate with the change in canonically spliced transcripts, suggesting that these other mechanisms do not play an appreciable role. It remains possible that RBM4 influences ULBP1 translation or gene expression in other cells or contexts.

It was recently reported that RBM4 regulates the alternative splicing of numerous cancer-related genes. Most interesting, in light of our findings, was the finding based on RNA-Seq analysis that RBM4 expression promoted a tumor-suppressive splicing profile (*Wang et al., 2014a*). This action occurred in part through the alternative splicing of the *BCL2L1* gene (*Bcl-x*), wherein RBM4 promoted splicing that generated the pro-apoptotic isoform Bcl-xS, at the expense of the anti-apoptotic isoform Bcl-xL. Our data suggest that RBM4 further promotes tumor suppression by favoring the canonical splicing of *ULBP1* mRNA, marking cells for elimination by the immune system. Decreased RBM4 expression has been observed in various cancers compared to paired healthy tissues, including lung, breast, and pancreatic cancers. Furthermore, lower RBM4 expression correlated with decreased patient survival (*Wang et al., 2014a*). We speculate that tumor cells are under selective pressure to downregulate RBM4 expression, as this would allow tumors to evade tumor-suppressive events such as expression of Bcl-xS and ULBP1.

An interesting question is whether the splicing functions of RBM4 are subject to stress-regulation. Intriguingly, the subcellular localization of RBM4 is reportedly regulated by oxidative stress or activation of the MKK$_{3/6}$-p38 MAPK pathway. In that instance, however, stress favored the cytoplasmic localization of RBM4 (*Lin et al., 2007*), where it reportedly enhanced translation of a subset of mRNAs (*Lin et al., 2007*; *Lin and Tarn, 2009*; *Uniacke et al., 2012*). Additional studies will be necessary to establish whether regulated re-localization of RBM4, or other types of regulation, impact alternative splicing of *ULBP1* or other transcripts.

## HSPA13

HSPA13 (also known as STCH) belongs to the Hsp70 family of chaperone proteins (*Otterson et al., 1994*). HSPA13 is associated with microsomes and displays ATPase activity, but little is known about its

function, including whether HSPA13 participates in cellular stress responses. One report showed that HSPA13 binds to the ion transporters NBCe1 and NHE1, which help maintain intracellular pH homeostasis (*Bae et al., 2013*). HSPA13 overexpression increased NBCe1 and NHE1 protein levels and functional activity. It will be interesting to pursue the mechanisms by which HSPA13 supports expression of ULBP1, which could be through direct or indirect effects.

## SPCS1 and SPCS2

SPCS1 and SPCS2 are non-catalytic subunits of the signal peptidase complex, but surprisingly little is known about their function (*Mullins et al., 1996*). We were surprised that HAP1 cells lacking *SPCS1* or *SPCS2* did not have a more general defect in cell surface protein expression; out of seven cell surface proteins tested, only ULBP1 expression was decreased. However, it should be noted that seven proteins represent a small fraction of the proteome, and we expect that a proteome-wide investigation would reveal many additional affected proteins. In yeast, SPCS1 and SPCS2 are required for maximal signal peptidase activity under certain growth conditions (*Fang et al., 1996*; *Mullins et al., 1996*), and in mammalian cells, SPCS1 supports expression of the HCV protein NS2 (*Suzuki et al., 2013*). It will be of future interest to define the properties that determine if a given secreted or membrane protein depends on SPCS1 and SPCS2 for maximal expression.

## Final remarks

The drivers of ULBP1 expression we have identified are likely to cooperate with previously described stress pathways that regulate ULBP1. Other transcription factors known to bind the *ULBP1* gene and drive gene expression include Sp1, Sp3, and the tumor suppressor protein p53 (*López-Soto et al., 2006*; *Textor et al., 2011*). It is possible that one or more of these factors contribute to *ULBP1* transcription in HAP1 and Jurkat cells, considering that residual ULBP1 expression was observed even after ATF4 was disrupted. Additional stress pathways including heat shock, E2F transcription factors, and the DNA damage response drive expression of other NKG2D ligands, and the combined expression level of all the NKG2D ligands on a given cell will contribute to its ability to activate NK cells (*Gasser et al., 2005*; *Venkataraman et al., 2007*; *Jung et al., 2012*). HAP1 cells may be representative of a very high 'threat level', as they had activated numerous ULBP1 drivers, and independent drivers for several other NKG2D ligands. Moreover, these cells possessed potent NK-activating properties distinct from NKG2D ligands, as shown by studies with NKG2D-deficient NK cells (data not shown). Probably as a consequence, loss of ULBP1 drivers, and even ULBP1 itself, had little detectable effect on the capacity of HAP1 cells to stimulate a response (data not shown).

The layers of ULBP1 regulation we have described could allow NKG2D ligand expression to be fine-tuned to reflect the 'threat level' of the cell. ATF4 could upregulate *ULBP1* transcription, but other factors may amplify the extent of ULBP1 transcription. Maximal cell surface expression would require RBM4 to direct splicing of the functional mRNA isoform, followed by maximal translation and trafficking of the protein to the cell surface. Ubiquitous RBM4 expression is 'safe' for normal, healthy cells, since functional *ULBP1* cannot be spliced unless the gene is transcribed in the first place.

The results of our screen highlight diverse mechanisms for the control of NKG2D ligand expression. To the best of our knowledge, none of the hits we investigated are related to previously known drivers of ULBP1 or other NKG2D ligands, and the effects found were generally specific to ULBP1. Screens for drivers of other NKG2D ligands will be the subject of future work, and the advent of CRISPR/Cas9-based screens will enable screens to be carried out in many different cell lines (*Gilbert et al., 2014*; *Shalem et al., 2014*; *Wang et al., 2014b*).

## Materials and methods

### Cells, antibodies, and reagents

Cell cultures were performed at 37°C in humidified atmosphere containing 5% $CO_2$. HAP1 cells were cultured in complete IMDM, consisting of IMDM (Life Technologies, Carlsbad, CA), 10% fetal calf serum (FCS, Omega Scientific, Tarzana, CA), 100 U/ml penicillin (Life Technologies), 100 µg/ml streptomycin (Life Technologies), and GlutaMAX-I (Life Technologies). K-562 and Jurkat cells were cultured in complete RPMI, consisting of RPMI (Life Technologies), 10% FCS, 1 mM sodium pyruvate (Life Technologies), MEM non-essential amino acids (Life Technologies), 100 U/ml penicillin, 100 µg/ml streptomycin, 0.2 mg/ml

glutamine (Sigma–Aldrich, St. Louis, MO), 10 µg/ml gentamycin sulfate (Lonza, Basel, Switzerland), 20 mM HEPES (Thermo Fisher Scientific, Waltham, MA), and 50 µM 2-mercaptoethanol (EMD Millipore, Billerica, MA).

Antibodies against ULBP1 (Clone 170818), ULBP2/5/6 (Clone 165903), ULBP3 (Clone 166510), MICA (Clone 159227), and MICB (Clone 236511) were purchased from R&D Systems. Antibodies against HLA-A,B,C (Clone W6/32), CD55 (Clone JS11), CD59 (Clone p282 H19), CD112 (Clone TX31), and CD155 (Clone TX24), and mouse Thy1.1 (Clone OX-7) were purchased from BioLegend (San Diego, CA). In some experiments, the antibodies against HLA-A,B,C (Clone W6/32) and CD59 (Clone OV9A2) were purchased from eBioscience (San Diego, CA). Antibodies against ATF4 used for chromatin immunoprecipitation (ChIP) were sc-200 (polyclonal) (Santa Cruz Biotechnology, Santa Cruz, CA), D4B8 (monoclonal) (Cell Signaling Technologies, Danvers, MA), and ABE387 (polyclonal) (EMD Millipore).

Histidinol, thapsigargin, and RNase A were purchased from Sigma–Aldrich. Proteinase K was purchased from Roche (Basel, Switzerland).

## Retroviral gene-trap mutagenesis and screen

Mutagenesis of HAP1 cells was performed as previously described (*Carette et al., 2011*). Selection of ULBP1$^{low}$ cells was performed using sequential rounds of selection. First, $10^8$ mutagenized HAP1 cells were labeled with ULBP1 antibody, and ULBP1$^{high}$ cells were depleted by running the cells over two sequential MACS LD columns (Miltenyi Biotec, San Diego, CA). Depletion of ULBP1$^{high}$ cells on this day was less efficient than we had achieved in previous experiments, so we chose to re-plate the selected cells, expand the cells for 2 days, and repeated the magnetic depletion of ULBP1$^{high}$ cells. We speculate that a single round of magnetic depletion may have been sufficient if adequate depletion of ULBP1$^{high}$ cells had been achieved. Magnetically selected cells were re-plated and expanded for 5 days. Expanded cells were labeled with ULBP1 and CD55 antibodies, and ULBP1$^{low}$CD55$^+$ cells were sorted by FACS, re-plated, and expanded. Genomic DNA was isolated from $4 \times 10^7$ cells. Mapping of gene-trap insertion sites and statistical analysis were performed as previously described (*Carette et al., 2011*).

## Flow cytometry

Cells were stained with the specified antibodies in 50 µl of PBS supplemented with 2.5% FCS or 1% BSA (FACS buffer). FACS buffer included 0.05% sodium azide for analytical flow cytometry, but not cell sorting. Dead cells were excluded from analysis by staining with DAPI (BioLegend). Multicolor flow cytometry was performed on an LSR Fortessa cytometer (BD Biosciences, San Jose, CA), and data were analyzed with FlowJo software (Tree Star, Inc., Ashland, Oregon).

## CRISPR/Cas9 gene targeting

Mutant HAP1 cells were generated by transiently co-transfecting cells with a Cas9 expression vector (pMJ918, a gift from Jennifer Doudna), an sgRNA expression vector (Addgene plasmid #41824, a gift from George Church), and a GFP expression vector using Lipofectamine 2000. Transfected cells were sorted based on GFP expression 24–72 hr post-transfection. To enrich cells with a mutant phenotype, cells were stained for ULBP1 expression 5–7 days post-transfection, and single ULBP1$^{low}$ cells were sorted into 96-well plates. Mutant cell clones were identified by sequencing PCR products surrounding the Cas9:sgRNA target site. sgRNA sequences are listed in *Table 2*. PCR primers used to detect mutations are listed in *Table 3*.

*ATF4* KO K-562, Jurkat, and HAP1 cells were generated using two sgRNAs flanking *ATF4* (ATF4 sgRNA 2 and ATF4 sgRNA 3) in order to delete the entire *ATF4* gene. Cells were transiently co-transfected with GFP and the Cas9/sgRNA expression plasmid px330 (Addgene plasmid #42230, a gift from Feng Zhang). Transfected cells were sorted based on GFP expression 24–72 hr post-transfection. 5–7 days post-transfection, single cells were sorted into 96-well plates without any additional selection.

## Lentiviral transduction of doxycycline-inducible constructs

For doxycycline-inducible gene expression, constructs encoding ATF4, RBM4, HSPA13, SPCS1, and SPCS2 were cloned into the lentiviral vector pFG12-TRE-UbC-rtTA-Thy1.1 (hereafter pFG12). The *ATF4* construct was a gift from Yihong Ye (Addgene plasmid #26114). The *RBM4* construct was a gift from Lan Ko, Georgia Health Sciences University. The *HSPA13* construct was a gift from Kyungpyo

**Table 2**. sgRNA sequences

| Name | sgRNA Sequence + PAM | Used in |
|---|---|---|
| ULBP1 sgRNA | **G**GGTCCCGGGCAGGATGGGT**CGG** | HAP1 |
| ATF4 sgRNA 1 | GGCGGGCTCCTCCGAATGGC**TGG** | HAP1 |
| ATF4 sgRNA 2 | **G**CTCGTCACAGCTACGCCCT**GGG** | HAP1, K-562, Jurkat |
| ATF4 sgRNA 3 | **G**TGGCCAACTATACGGCTCCA**GGG** | HAP1, K-562, Jurkat |
| RBM4 sgRNA | **G**AGTCCCACCTGCACCAATA**AGG** | HAP1 |
| RBM4/RBM4b sgRNA | **G**CCCCGGGAGGCTACAGAGC**AGG** | HAP1 |
| HSPA13 sgRNA | GTGCCAAATAGCCGGCCAAC**AGG** | HAP1 |
| SPCS1 sgRNA | **G**CATCTGTTCAGCTAGCTTC**TGG** | HAP1 |
| SPCS2 sgRNA | **G**AGTGGCCGTAGCGGCTTGT**TGG** | HAP1 |

PAM, protospacer adjacent motif.
For each sgRNA, the protospacer adjacent motif (PAM) is indicated in bold blue text. Red text indicates that the 5' G is not present in the genomic target sequence and was added to the sgRNA to allow transcription from the U6 promoter.

Park, Seoul National University. The *SPCS1* construct was a gift from Tetsuro Suzuki, Hamamatsu University School of Medicine. *SPCS2* was cloned from HAP1 cDNA.

Lentiviral supernatants were generated by co-transfecting 293T cells with pFG12 vector, the VSV-G plasmid pMD2.G, and the packaging plasmid psPAX2 using Lipofectamine 2000 (Life Technologies). Supernatants were harvested 48 hr post-transfection, filtered with a 0.45-µM PES syringe filter, and added to target cell cultures with 4 µg/ml polybrene. Transduced cells were sorted based on Thy1.1 expression. To induce gene expression, cells were cultured in complete medium with 100 or 1000 ng/mL doxycycline (Sigma–Aldrich), as noted in figure legends.

## Quantitative real-time PCR
Total RNA was isolated from cells using the RNeasy Mini Kit and on-column DNase digestion using the RNase-free DNase Set (Qiagen, Hilden, Germany). Reverse transcription was performed with iScript

**Table 3**. PCR primers used to detect mutations

| Target | Primer direction | Primer sequence | Amplicon size (bp) |
|---|---|---|---|
| ULBP1 sgRNA target locus | Forward | ATAAACAGCCGTGGTGTGAG | 401 |
| | Reverse | TGTCTGGGGAGATCACGATG | |
| ATF4 sgRNA 1 target locus | Forward | CATTCCTCGATTCCAGCAAAGC | 345 |
| | Reverse | TGAGTGATGGGGCCAAGTGAG | |
| Deletion of *ATF4* with sgRNAs 2 and 3 | Forward | CGTCCTCGGCCTTCACAATA | 442 |
| | Reverse | TCTTCAGGATGAGGCTTCTGC | |
| RBM4 sgRNA target locus | Forward | TGCACATAGAAGACAAGACGGC | 332 |
| | Reverse | CCTTGTGTTCAGCCCTCTACCC | |
| HSPA13 sgRNA target locus | Forward | TGCTGTCTGAGAGGAGTGCT | 476 |
| | Reverse | CCTCCAACTCTTCTGCGGTA | |
| SPCS1 sgRNA target locus | Forward | ATTTAATATCTTGCCCAGGCCC | 394 |
| | Reverse | ACCCACAAATTTCTTACCAAACAT | |
| SPCS2 sgRNA target locus | Forward | AACCTCAAGTCCCAGCAAGC | 466 |
| | Reverse | CGGGTCCAGGTTTGAAGTGT | |

Reverse Transcription Supermix (Bio-Rad, Hercules, CA). Quantitative real-time PCR was performed on a CFX96 thermocycler using SsoFast EvaGreen Supermix (Bio-Rad) with the following cycling program: Initial denaturation at 98°C, 2′, followed by 40 cycles of 98°C, 2″, 55°C, 5″. Data were analyzed with Bio-Rad CFX Manager Software version 3.1. Gene expression was normalized to the reference genes *ACTB*, *GAPDH*, and *HPRT1*, normalizing the gene of interest to the geometric mean of the three reference genes (*Vandesompele et al., 2002*). qPCR primer sequences are listed in *Table 4*.

## ChIP and ChIP-Seq

Cells were re-plated in fresh media 12–24 hr prior to treatment. Media was then replaced with fresh media or fresh media containing 2 mM histidinol, and cells were incubated for 24 hr. Chromatin

**Table 4**. Primers used for qPCR

| Target | Primer direction | Primer sequence (5′-3′) | Amplicon size (bp) |
|---|---|---|---|
| ACTB | Forward | TTGGCAATGAGCGGTTCC | 92 |
| | Reverse | GTTGAAGGTAGTTTCGTGGATG | |
| GAPDH | Forward | CAACAGCGACACCCACTCCT | 115 |
| | Reverse | CACCCTGTTGCTGTAGCCAAA | |
| HPRT1 | Forward | AGGATTTGGAAAGGGTGTTTATTC | 109 |
| | Reverse | CAGAGGGCTACAATGTGATGG | |
| Total ULBP1 (exon 3-exon 3) | Forward | GCCAGGATGTCTTGTGAGCATGAA | 134 |
| | Reverse | TTCTTGGCTCCAGGATGAAGTGCT | |
| ULBP1 canonical isoform (exon 1-exon 2) | Forward | ATCAGCGCCTCCTGTCCAC | 136 |
| | Reverse | AAAGACAGTGTGTGTCGACCCAT | |
| ULBP1 alternative isoform (extended exon 1-exon 2) | Forward | GGAATTGCAGGAGGGTGGAG | 183 |
| | Reverse | CAAAGGCTTTGGCCTTGTGGTTAA | |
| ULBP1 spliced transcript (exon 2-exon 3) | Forward | TAAGTCCAGACCTGAACCACA | 477 |
| | Reverse | CCATTGAAGAGGAACTGCCAAG | |
| ULBP1 alternative isoform and primary transcript (exon 1-extended exon 1) | Forward | CCGGGCAGGATGGGTCG | 263 |
| | Reverse | TGTCTGGGGAGATCACGATG | |
| ULBP1 unspliced transcript (Intron 1-exon 2) | Forward | CCCTCAGAGGCCTTCACTTG | 195 |
| | Reverse | AAGGCCTTTCATCCACCAGG | |
| ULBP2 | Forward | GCCGCTACCAAGATCCTTCT | 161 |
| | Reverse | TCATCCACCTGGCCTTGAAC | |
| ULBP3 | Forward | CTCGCGATTCTTCCGTACCT | 127 |
| | Reverse | TCTGGACCTCACACCACTGT | |
| MICA | Forward | ATGTCCTGCCTGATGGGAATGGAA | 189 |
| | Reverse | CAGCAGCAACAGCAGAAACATGGA | |
| MICB | Forward | TGGATCTGTGCAGTCAGGGTTTCT | 176 |
| | Reverse | TGAGGTCTTGCCCATTCTCTGTCA | |
| ULBP1 promoter ChIP | Forward | GCTGTCAGATGACGAGCCC | 80 |
| | Reverse | ATACACTGGGCGGGATCCTA | |
| ASNS promoter ChIP | Forward | TGGTTGGTCCTCGCAGGCAT | 66 |
| | Reverse | CGCTTATACCGACCTGGCTCCT | |
| ASNS exon 7 ChIP | Forward | GCAGCTGAAAGAAGCCCAAGT | 62 |
| | Reverse | TGTCTTCCATGCCAATTGCA | |

complexes were cross-linked by replacing the culture media with fresh complete media containing 1% formaldehyde, and cells were incubated at room temperature for 10 min. Cross-linking was quenched by adding glycine to a final concentration of 0.125 M. Cells were washed two times with ice-cold PBS, and cell pellets were flash-frozen in a slurry of dry ice and ethanol.

Cells were resuspended in ChIP Lysis Buffer #1 (see buffer compositions below), incubated for 10′ at 4°C with rotation, and centrifuged for 8′ at 800×g at 4°C. Pellets were resuspended in ChIP Lysis Buffer #2, incubated for 10′ at 4°C with rotation, and centrifuged for 8′ at 800×g at 4°C. Pellets were resuspended in 3 volumes (~200 µl) of ChIP Lysis Buffer #3. Chromatin was sheared to an average fragment size of 200 bp using a Covaris S2 Ultrasonicator (Covaris, Woburn, MA). Sheared chromatin was diluted in additional ChIP Lysis Buffer #3, followed by addition of Triton X-100 (final concentration 1%) and NaCl (final concentration 150 mM) and divided into 1 ml aliquots for immunoprecipitation. Antibody was added to chromatin samples, followed by overnight incubation at 4°C, with rotation. Antibody-chromatin complexes were captured with Protein G Dynabeads (Life Technologies) at 4°C for 2 hr, with rotation. Beads were washed 2× with low-salt wash buffer, 1× with high-salt wash buffer, 1× with LiCl wash buffer, and 2× with TE. Beads were resuspended in ChIP elution buffer, and chromatin was eluted by incubating samples at 65°C for 15′. Supernatant containing eluted chromatin was removed from the beads and the cross links were reversed by incubating samples overnight at 65°C.

RNase A was added to samples to a final concentration of 0.2 mg/ml, followed by incubation at 37°C for 1 hr. Proteinase K was then added to a final concentration of 0.4 mg/ml, followed by incubation at 56°C for 1 hr. DNA was then isolated using the Qiagen MinElute Kit, followed by qPCR as described above or Illumina Library prep.

ChIP-Seq libraries were prepared using the NEBNext Ultra DNA Library Prep Kit for Illumina (New England BioLabs, Ipswich, MA), including optional size selection of adapter-ligated DNA and 13 cycles of PCR amplification. Final libraries were analyzed with a Qubit fluorometer (Life Technologies) and Bioanalyzer 2100 (Agilent Technologies, Santa Clara, CA) to assess quantity and quality. Libraries were clustered at a density of 6.5 pM with a cluster station and sequenced on a GAIIx Genome Analyzer (Illumina, San Francisco, CA).

ChIP-Seq reads were aligned to the human genome (hg19) using Bowtie2 version 2.1.0 using all default settings (*Langmead and Salzberg, 2012*). Reads that could not be uniquely mapped were discarded. Read densities were normalized to the total number of aligned reads in a sample using Bedtools version 2.17.0 (*Quinlan and Hall, 2010*). ATF4-binding peaks were identified with HOMER version 4.7 (*Heinz et al., 2010*) using the program findPeaks with the following options:

-style factor -i <input_tagdir>

where <input_tagdir> is a directory of ChIP-Seq reads from the input DNA fraction. ATF4 binding motifs were identified with HOMER version 4.7 using the program findMotifsGenome.pl with the following options:

-size 50 -mask

Data were visualized with the Integrative Genomics Viewer (IGV) version 2.3.47 (*Robinson et al., 2011*).

## ChIP buffer compositions and antibody concentrations

ChIP Lysis Buffer #1: 50 mM HEPES pH 7.5, 140 mM NaCl, 1 mM EDTA, 10% glycerol, 0.5% NP-40, 0.25% Triton X-100, Protease Inhibitors*

ChIP Lysis Buffer #2: 10 mM Tris pH 8.0, 200 mM NaCl, 1 mM EDTA, 0.5 mM EGTA, Protease Inhibitors*

ChIP Lysis Buffer #3: 10 mM Tris pH 8.0, 100 mM NaCl, 1 mM EDTA, 0.5 mM EGTA, 0.1% sodium deoxycholate, 0.5% N-lauroylsarcosine, Protease Inhibitors*

Low-salt wash buffer: 20 mM Tris pH 8.1, 150 mM NaCl, 2 mM EDTA, 0.1% Sodium dodecyl sulfate (SDS), 1% Triton X-100

High-salt wash buffer: 20 mM Tris pH 8.1, 500 mM NaCl, 2 mM EDTA, 0.1% SDS, 1% Triton X-100

LiCl salt wash buffer: 20 mM Tris pH 8.1, 0.25 M LiCl, 1 mM EDTA, 1% NP-40, 1% sodium deoxycholate

TE: 10 mM Tris pH 8.0, 1 mM EDTA

Elution buffer: 10 mM Tris pH 8.0, 300 mM NaCl, 1 mM EDTA, 1% SDS

*Complete Mini EDTA-free protease inhibitor cocktail (Roche, Basel, Switzerland)

Antibody amounts per 1 ml IP: anti-ATF4 (sc-200): 5 µg, anti-ATF4 (D4B8): 1.06 µg, anti-ATF4 (ABE387): 20 µg, Rabbit IgG control: 5 µg

## RNA-Seq

Total RNA was purified with RNAzol RT (Sigma–Aldrich) according to the manufacturer's instructions. For *RBM4* KO samples, equal amounts of total RNA were pooled from 4 independent *RBM4* KO cell clones. For *ATF4* KO samples, equal amounts of total RNA were pooled from three independent *ATF4* KO cell clones. RNA-Seq libraries were prepared using the NEBNext Ultra Directional RNA Library Prep Kit (New England BioLabs), including poly(A) selection using the NEBNext Poly(A) mRNA Magnetic Isolation module. Library quality assessment and clustering were performed as described for ChIP-Seq. Paired-end sequencing was performed using a GAIIx Genome Analyzer (Illumina).

RNA-Seq reads were aligned using Tophat version 2.0.11 (*Kim et al., 2013*) using the reference human genome hg19 and the reference transcriptome CRCh37.59 using the following options:

–library-type fr-firststrand –no-coverage-search –mate-inner-dist 300

Guided transcriptome reconstruction was done with Cufflinks Version 2.2.1 (*Trapnell et al., 2012*). Cufflinks was run on the Tophat output file 'accepted_hits.bam' for each sample, using the hg19 reference genome and transcript annotation files and the options

-b hg19_ucsc.fa -M hg19.rRNA.gtf -g hg19_ucsc.refGene.gtf -u –min-frags-per-transfrag

After assembly was complete for each sample, the assembled transcripts were merged together using Cuffmerge with the command template:

cuffmerge -o combined/ –min-isoform-fraction 0 -s hg19 assembly_list.txt

The merged transcriptome was compared against the reference transcriptome using Cuffcompare: cuffcompare -r hg19_ucsc.refGene.gtf -s hg19 -R -M merged.gtf

This produced comparative statistics, and most importantly for this pipeline, a '.tmap' file that lists the most closely matching reference transcript for each of the assembled transcripts from merged.gtf. The merged transcriptome was converted to genePred format (.gp) using the UCSC binary utility, gtfToGenePred using the command:

gtfToGenePred merged.gtf merged.gp

merged.gp was joined to the .tmap file produced by Cuffcompare above, to obtain a transcriptome file that is annotated with transcript IDs, exonic coordinates, gene IDs, and short-names of the most closely matching reference transcripts. The Tophat output files 'accepted_hits.bam' for each sample were sorted and converted to BED format using the BEDTools-2.17.0 utility bamToBed with the command:

bamToBed -split -i accepted_hits.bam|sort -k1,1 -k2,2n -k3,3n > accepted_hits.bed

An executable (written in-house) called kcGEXM (*Escobar et al., 2014*) was used to obtain normalized read counts and calculate reads per kilobase per million (RPKM) for transcripts. kcGEXM uses the genePred and BED files as input:

kcGEXM -f gp -r merged.tmap.gp accepted_hits.bed > sample_name.cnt

where -f precedes the format of the reference annotation file and -r precedes the reference annotation file.

In *Figure 5—figure supplement 1*, read densities were normalized to the total number of aligned reads in a sample using Bedtools version 2.17.0 (*Quinlan and Hall, 2010*), and data were visualized with IGV version 2.3.47 (*Robinson et al., 2011*). Sashimi plots were generated using MISO version 0.5.2 (*Katz et al., 2010*) with Python version 2.7.

## Luciferase reporter assay

Fragments of the *ULBP1* promoter were amplified from HAP1 genomic DNA and inserted into the pGL3-Basic luciferase expression vector (Promega, Fitchburg, WI) between the KpnI and HindIII restriction sites. Mutant *ULBP1* promoter fragments were ordered from IDT (Coralville, IA). The sequences of the *ULBP1* promoter constructs can be found in *Supplementary files 2–7*. *ATF4*-mutant HAP1 cells were plated at 10,000 cells/well in 96-well plates. 24 hr after plating, cells were co-transfected with the indicated *ULBP1* promoter constructs (99 ng), pRK5-FLAG-ATF4 or control vector (199 ng), and renilla luciferase vector pRL-SV40 (1 ng). 24 hr post-transfection, cells were washed twice in PBS and lysed with 50 µl of passive lysis buffer (Promega #E1941). 15 µl of lysate was transferred to an opaque 96-well assay plate. 100 µl of D-Luciferin reagent (synthesized in-house) was added to the lysates and samples were immediately assessed for luminescence over a 10-s time period using an LMAX-II luminometer (Molecular Devices, Sunnyvale, CA).

## Acknowledgements

We thank all members of the Raulet lab for helpful discussion and comments on the manuscript. We thank Hector Nolla, Alma Valeros, and Kartoosh Heydari for help with cell sorting; Michele Ardolino, Troy Trevino, Josh Johnson, and Lily Zhang for technical assistance; and George Church, Lan Ko, Kyungpyo Park, Tetsuro Suzuki, Yihong Ye, and Feng Zhang for expression constructs. We thank Jennifer Doudna for advice and the Cas9 expression construct; Michael Kilberg and Kaoru Saijo for reagents and advice concerning ATF4 ChIP, Chrysi Kanellopoulou for the ChIP-seq protocol, Chris Benner for advice with HOMER, and the NIAID high performance computing cluster. We thank Cuong K Nguyen, Gokhul Kilaru, and Steven Witte for assistance with computational analyses. Deep sequencing of DNA for the retroviral gene-trap screen was performed at the Stanford Functional Genomics Facility.

## Additional information

### Funding

| Funder | Grant reference | Author |
|---|---|---|
| National Institutes of Health (NIH) | Intramural Research Program of the National Institute of Allergy and Infectious Diseases | Stefan A Muljo |
| National Science Foundation (NSF) | Research Experience for Undergraduates, NSF 0552996 | Teal Russell |
| David and Lucile Packard Foundation | Fellowship | Jan E Carette |
| University of California Berkeley (UC Berkeley) | K. Peter Hirth Chair, Graduate Student Fellowship | Benjamin G Gowen |
| National Institutes of Health (NIH) | R01-AI039642 | David H Raulet |
| National Institutes of Health (NIH) | DP2 AI104557 | Jan E Carette |
| National Science Foundation (NSF) | Graduate Research Fellowship DGE 1106400 | Benjamin G Gowen |

The funders had no role in study design, data collection and interpretation, or the decision to submit the work for publication.

### Author contributions

BGG, JEC, Conception and design, Acquisition of data, Analysis and interpretation of data, Drafting or revising the article; BC, TTG, Acquisition of data, Analysis and interpretation of data, Drafting or revising the article; CDM, PB, JRG, CRH, PAD, TR, AI, Acquisition of data; LC, Conception and design, Analysis and interpretation of data; CLS, Contributed unpublished essential data or reagents; SAM, DHR, Conception and design, Analysis and interpretation of data, Drafting or revising the article

## Additional files

### Supplementary files

• Supplementary file 1. Complete list of hits and statistical analysis of all independent insertions mapped in the ULBP1 screen data set. (A) Annotated list of hits: The gene symbols of hits (p < 0.05) with a brief description of known or predicted gene functions. A p-value of enrichment was determined using Fisher's exact test, followed by correction for the false discovery rate. Yellow shading indicates genes confirmed in this study to impact ULBP1 expression. Light blue shading indicates genes involved in GPI biosynthesis and anchoring. Light orange shading genes involved in protein glycosylation. Gray shading indicates genes that have occurred in several unrelated screens using the same cells, perhaps indicating pleiotropic effects. Asterisks indicate two genes (*SLC17A9* and *CRNKL1*) that, when targeted with CRISPR/Cas9, failed to result in decreased ULBP1 expression.

(B) Statistical analysis: Statistical analysis of all independent insertions mapped in the ULBP1 screen data set.

• Supplementary file 2. Sequence of ULBP1 promoter –603 construct. Annotated plasmid sequences of the ULBP1 promoter reporter constructs used in *Figure 6—figure supplement 2*.

• Supplementary file 3. Sequence of ULBP1 promoter –288 construct. Annotated plasmid sequences of the ULBP1 promoter reporter constructs used in *Figure 6—figure supplement 2*.

• Supplementary file 4. Sequence of ULBP1 promoter m1 construct. Annotated plasmid sequences of the ULBP1 promoter reporter constructs used in *Figure 6—figure supplement 2*.

• Supplementary file 5. Sequence of ULBP1 promoter m2 construct. Annotated plasmid sequences of the ULBP1 promoter reporter constructs used in *Figure 6—figure supplement 2*.

• Supplementary file 6. Sequence of ULBP1 promoter m3 construct. Annotated plasmid sequences of the ULBP1 promoter reporter constructs used in *Figure 6—figure supplement 2*.

• Supplementary file 7. Sequence of ULBP1 promoter m4 construct. Annotated plasmid sequences of the ULBP1 promoter reporter constructs used in *Figure 6—figure supplement 2*.

## Major datasets

The following datasets were generated:

| Author(s) | Year | Dataset title | Dataset ID and/or URL | Database, license, and accessibility information |
|---|---|---|---|---|
| Chim B, Gowen BG, Muljo SA, Raulet DH | 2015 | ULBP1_alternative_mRNA | http://www.ncbi.nlm.nih.gov/nuccore/KT591165 | Publicly available at GenBank (Accession number KT591165). |
| Carette J | 2015 | HAP1 gene trap unselected control dataset | http://www.ncbi.nlm.nih.gov/biosample/3703230 | Publicly available at the NCBI BioProject database (Accession no: SAMN03703230). |
| Carette J | 2015 | HAP1 gene trap ULBP1 screen | http://www.ncbi.nlm.nih.gov/biosample/3703231 | Publicly available at the NCBI BioProject database (Accession no: SAMN03703231). |
| Muljo SA, Raulet DH, Gowen BG, Chim B, Burr P | 2015 | Genome-wide analysis of ATF4 binding in histidinol-treated cells | http://www.ncbi.nlm.nih.gov/geo/query/acc.cgi?acc=GSE69304 | Publicly available at the NCBI Gene Expression Omnibus (Accession no: GSE69304). |
| Muljo SA, Raulet DH, Gowen BG, Chim B, Burr P | 2015 | RNA-Seq experiment to identify genes regulated by ATF4 | http://www.ncbi.nlm.nih.gov/geo/query/acc.cgi?acc=GSE69308 | Publicly available at the NCBI Gene Expression Omnibus (Accession no: GSE69308). |

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
