## [Decision Letter]

Thank you for submitting your work entitled “A forward genetic screen reveals novel independent regulators of ULBP1, an activating ligand for natural killer cells” for peer review at *eLife*. Your submission has been favorably evaluated by Tadatsugu Taniguchi (Senior Editor) and three reviewers, one of whom is a member of our Board of Reviewing Editors.

The reviewers have discussed the reviews with one another and the Reviewing Editor has drafted this decision to help you prepare a revised submission.

The manuscript by Gowen et al. reports on a retroviral trap-based genetic screen in the human haploid cancer cell line HAP1 to identify regulators of ULBP1, a ligand for NKGD2. They identify several hits that are confirmed by CRISPR/Cas9 gene inactivation. The major emphasis of the manuscript is on ATF4, which is shown to positively regulate ULBP1 in two other cancer cell lines. This effect is accentuated by activation of the unfolded protein response, and is relatively selective for ULBP1 in comparison to other NKGD2 ligands. Consistent with this, ATF4 is shown to bind near the *ULBP1* promoter, but not near genes encoding other NKGD2 ligands. The authors further show that a second hit, RBM4, is required for appropriate splicing of ULBP1. The authors conclude that their findings provide insights into the stress pathways that alert the immune system to danger. Overall the experiments appear to be carefully performed and clearly establish ATF4 and RBM4 as regulators of the expression ULBP1 in the cell lines examined. There was substantial enthusiasm for these findings among the three reviewers. One reviewer remarked: “Overall this paper provides new information on the regulation of NKG2D ligands expression, identifying 5 molecules previously unknown to be involved in ULBP1 expression. The method used is elegant, the results are convincing and the paper is well written.” A second reviewer commented: “Overall this is an excellent and insightful study with novel findings that are fully supported by the data and importantly advance the field on NKG2D/NKG2DL research.”

1) One significant concern was raised that needs to be addressed in a revised manuscript. The authors conclude that binding of ATF4 to the *ULBP1* gene indicates direct transcriptional control. That is almost certainly the case given the other evidence, but DNA binding per se is not a sufficient proof of direct regulation. Many transcription factor binding events are not associated with gene activation when assessed on a genome-wide scale. An assessment of the requirement of the binding site for activation is needed. This could be done fairly easily by a transient reporter gene assays using wild type and ATF4 binding site mutants.

2) The RBM4 mutation seems to affect MICA transcription in the first intron. The authors may comment on that.

---

## [Author Response]

*1) One significant concern was raised that needs to be addressed in a revised manuscript. The authors conclude that binding of ATF4 to the* ULBP1 *gene indicates direct transcriptional control. That is almost certainly the case given the other evidence, but DNA binding per se is not a sufficient proof of direct regulation. Many transcription factor binding events are not associated with gene activation when assessed on a genome-wide scale. An assessment of the requirement of the binding site for activation is needed. This could be done fairly easily by a transient reporter gene assays using wild type and ATF4 binding site mutants*.

Figure 6—figure supplement 2 has been added to address this concern. The data from transient luciferase reporter assays indicate that a fragment of the *ULBP1* promoter is activated by ATF4. Mutation of the ATF4 binding motif found in the *ULBP1* promoter ablates this response. Together with the other data in the paper, these data provide solid evidence that ATF4 drives transcription of the *ULBP1* gene.

*2) The RBM4 mutation seems to affect MICA transcription in the first intron. The authors may comment on that*.

In Figure 7—figure supplement 1, we did not observe any significant change in the number of reads in the first intron. On the other hand, the *RBM4* KO sample does have more RNA-seq reads spanning the splice junction between exons 1 and 2 than does the WT sample. The meaning of this result is not clear, especially considering that Figure 3 shows no difference in MICA surface expression between WT and *RBM4* KO cells. We think it is best to not comment on this in the main text and merely present the data as it is in the supplement, with a short comment in the figure legend.